# Insights into receptor structure and dynamics at the surface of living cells

**Frederik Steiert** [1,2,6]**, Peter Schultz**[1,6]**, Siegfried Höfinger**[3,4]**, Thomas D. Müller**[5]**, Petra Schwille** [1] **& Thomas Weidemann** [1]✉

Evaluating protein structures in living cells remains a challenge. Here, we investigate Interleukin-4 receptor alpha (IL-4Rα) into which the non-canonical amino acid bicyclo[6.1.0]nonyne-lysine (BCNK) is incorporated by genetic code expansion. Bioorthogonal click labeling is performed with tetrazine-conjugated dyes. To quantify the reaction yield in situ, we develop brightness-calibrated ratiometric imaging, a protocol where fluorescent signals in confocal multi-color images are ascribed to local concentrations. Screening receptor mutants bearing BCNK in the extracellular domain uncovered site-specific variations of both click efficiency and Interleukin-4 binding affinity, indicating subtle well-defined structural perturbations. Molecular dynamics and continuum electrostatics calculations suggest solvent polarization to determine site-specific variations of BCNK reactivity. Strikingly, signatures of differential click efficiency, measured for IL-4Rα in ligand-bound and free form, mirror sub-angstrom deformations of the protein backbone at corresponding locations. Thus, click efficiency by itself represents a remarkably informative readout linked to protein structure and dynamics in the native plasma membrane.

Site-specific manipulation of proteins by genetic code expansion (GCE), a technique that introduces non-canonical amino acids (ncAA) into the otherwise native polypeptide chain, offers fascinating possibilities for the investigation of protein function in living cells[1,2]. The introduced ncAA carries bioorthogonal reactivity for the site-specific conjugation of biophysical probes featuring negligible interference with components of the cell[3,4]. A variety of chemical strategies termed click labeling emerged for site-specific conjugation of sensor molecules to ncAA under physiological conditions[5]. As an alternative to metal-catalyzed click reactions, inverse electron-demand Diels-Alder cycloaddition (iEDDAC) of tetrazine derivatives to ncAA side chains presenting reactive dienophiles provides superior reaction rates for fast coupling to targets expressed via GCE[6–11]. Bioorthogonal click labeling with fluorescent dyes was successfully combined with biophysical single-cell methods to address common issues in receptor research, like diffusion, oligomerization states, and conformational changes[12–18]. However, only few attempts have been made to systematically quantify the yields of bioorthogonal reactions per se[19]. Because the experimental throughput is limited, efforts to optimize click reaction conditions can be substantial[20,21].

Fluorescence fluctuation analysis provides a simple means for transforming confocal images into molecular concentration maps[22] and is therefore a powerful extension for live cell imaging[23,24]. Consequently, fluorescence fluctuation analysis was introduced as a routine validation protocol in bioorthogonal labeling[25]. However, even elaborate instrumentation cannot circumvent the systematic errors inherent to single-cell measurements. Cells are far-from-equilibrium systems that carry out a myriad of processes on various time scales. Data curation must be carefully executed post-measurement to extract meaningful information. To mitigate these efforts, we here develop

[1]Department of Cellular and Molecular Biophysics, Max Planck Institute of Biochemistry, Am Klopferspitz 18, 82152 Martinsried, Germany. [2]Department of Physics, Technical University Munich, 85748 Garching, Germany. [3]VSC Research Center, TU Wien, Operngasse 11 / E057-09, 1040 Wien, Austria. [4]Department of Physics, Michigan Technological University, 1400 Townsend Drive, 49931 Houghton, MI, USA. [5]Biozentrum, Julius-von-Sachs-Institut für Biowissenschaften, Lehrstuhl für Molekulare Pflanzenphysiologie und Biophysik - Botanik I, Julius-von-Sachs-Platz 2, 97082 Würzburg, Germany. [6]These authors contributed equally: Frederik Steiert, Peter Schultz. ✉e-mail: weidemann@biochem.mpg.de

brightness-calibrated ratiometric imaging (BCRI), a protocol that strictly separates molecular brightness analysis from routine dual-color imaging of labeled cells. We demonstrate that BCRI allows for accelerated screening of a broad range of labeling conditions at the cell surface with moderate sample sizes and greatly improved reproducibility.

Bioorthogonal labeling is particularly suitable for protein domains which are accessible from bulk solution. As a model system we chose the single-pass transmembrane subunit Interleukin-4 receptor alpha (IL-4Rα), for which a wealth of information is available: a comprehensive set of crystal structures[26,27], a precisely mapped extracellular ligand binding epitope[28,29], and dynamic behavior in the plasma membrane of living cells[30–33]. Employing BCRI, we characterize cellular labeling of ncAA incorporated at different positions of extracellular IL-4Rα domains. Supported by molecular dynamics (MD) simulations, we can link site-specific variations of the click reaction to physico-chemical as well as structural properties of the target protein. The single-amino acid resolution of click labeling and the robust quantitative readout gained by BCRI bears unexpected potential to address protein structure and dynamics at full atomistic detail in living cells.

## Results

### Multi-color labeling of receptors

To study bioorthogonal labeling of a surface-expressed protein, we used a truncated, non-signaling IL-4Rα construct transiently expressed in HEK293T cells, henceforth referred to as IL-4Rα*[30,34]. IL-4Rα* carries a hexahistidine stretch (His-tag) at the extracellular N-terminus and an intracellular C-terminal enhanced green fluorescent protein (eGFP) domain. The ncAA bicyclo[6.1.0]nonyne-lysine (BCNK) was inserted by overriding a stop-codon (TAG) in the coding sequence with BCNK-

loaded tRNA$^{UAG}$. For loading, modified pyrrolysyl-tRNA synthethase and four copies of tRNA$^{UAG}$ were co-expressed from a concomitant plasmid[35,36]. We selected 14 solvent-exposed side chains in the extracellular domain for BCNK replacement, henceforth referred to as receptor mutants (Fig. 1a). In accordance with IL-4Rα*, the trafficking rates of the receptor mutants led to accumulation in the plasma membrane, which facilitates the quantification of cell surface-associated signals. The expression system was tight as cells lacking any component of the GCE-toolkit showed negligible fluorescence. Titrating BCNK in the growth medium, the expression levels followed a logistic curve with a plateau at about 500 µM (Supplementary Fig. 1 and Supplementary Table 1). In contrast, increasing expression levels of synthetase and tRNA had no effect. Interestingly, the transfection efficiencies varied strongly among the receptor mutants (Supplementary Fig. 2 and Supplementary Table 2).

BCNK-bearing receptor mutants at the surface of living cells were labeled with tetrazine-functionalized fluorescent dyes via iEDDAC (Supplementary Fig. 3). For functional comparison, we designed several reference channels (Fig. 1b). The total pool of receptors was quantified by eGFP signal. Reversible binding of trisNTA-Alexa647[30] and fluorescently labeled IL-4[37] were used to probe for accessibility and structural integrity of the receptor mutants. For example, after click labeling with tetrazine-functionalized Alexa568 (Alexa568-tet1), occupied receptors colocalize all three labels at the plasma membrane but not inside the cell (Fig. 1c). However, since signals from different color channels are unrelated, quantitative molecular relations like the fraction of labeled receptors cannot be deduced from these images.

To gain molecular insight, we first applied dual-color fluorescence cross-correlation spectroscopy (FCCS) (Supplementary Fig. 4). Based on the eGFP channel, the diffusion coefficients of $0.16 \pm 0.05$ µm$^2$s$^{-1}$ for

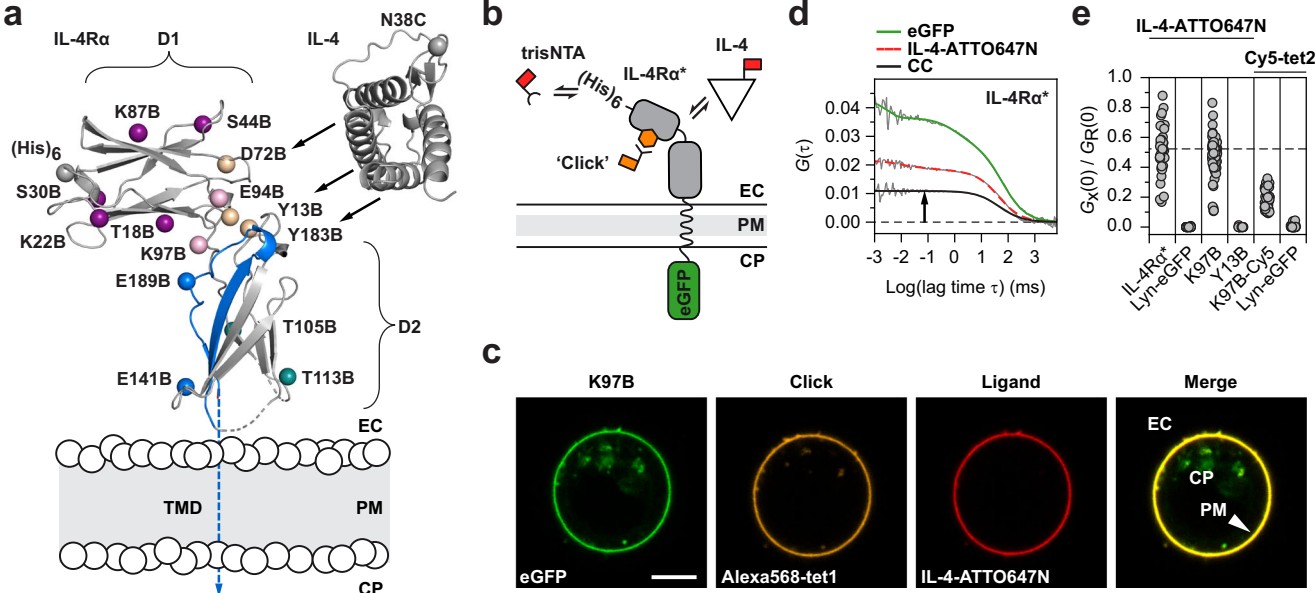

**Fig. 1 | Live-cell labeling of receptors with multiple ligands. a** Structure of IL-4Rα ectodomains (D1 and D2; PDB: 1IAR, backbone in 'cartoon' representation) and natural ligand IL-4 aligned for binding (black arrows) placed to scale on a schematic plasma membrane (PM; EC, extracellular; CP, cytoplasmic compartments). BCNK incorporation (indicated by B) sites (Cβ atoms in 'spheres' representation) of the receptor mutants are indicated by mature residue numbers (one-letter-code; Uniprot: P24394) and color-coded by domain: D1 (purple), the linker between D1 and D2 (pink), the activation loop in D2 (blue), D2 sites distant from the activation loop (teal), and neutralizing sites at the IL-4 binding interface (wheat). His-tag at the N-terminus ((His)₆, gray 'sphere'), and maleimide-coupled ATTO647N within IL-4 (N38C, gray 'sphere') serve as functional labels. The PM is traversed by a single transmembrane domain (TMD, dashed blue arrow). **b** Schematic representation of

reference channels to validate functional subpopulations of labeled receptors: expression (eGFP), surface accessibility (trisNTA), and structural integrity (IL-4). **c** Representative equatorial confocal cross-sections showing multi-labeled receptors at the plasma membrane of a single HEK293T cell. Scale bar: 5 µm. **d** Example correlation functions with a finite cross-correlation amplitude (black arrow) indicating co-diffusion of IL-4Rα* and IL-4-ATTO647N. **e** Ratio of cross- and auto-correlation amplitudes reflecting the fraction of receptors occupied with IL-4-ATTO647N or clicked with Cy5-tet2 relative to IL-4Rα* (dashed line). Receptor mutants K97B and Y13B contain BCNK, K97B-Cy5 represents receptor mutant K97B clicked with Cy5-tet2, and Lyn-eGFP serves as plasma membrane marker (illustrated in Supplementary Fig. 4a). Experiments were performed once (Lyn-eGFP, Y13B) or twice (else). Source data are provided as a Source Data file.

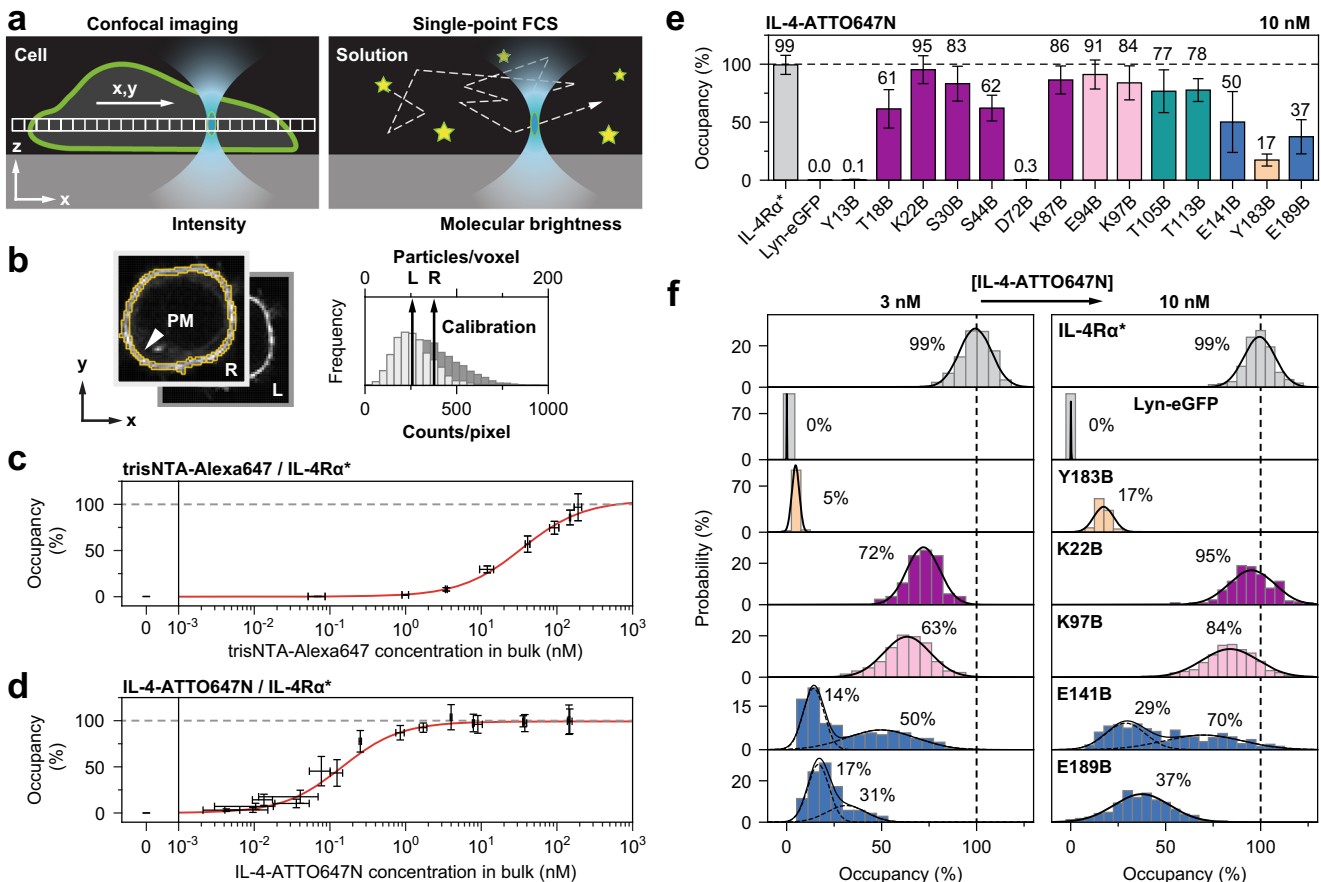

**Fig. 2 | Brightness-calibrated ratiometric imaging (BCRI). a** Schematic side view of the confocal detection volume (cyan) during raster scanning (arrow) of a cell and static positioning in free solution. Stochastic movements (dashed line) of the fluorescent particles (stars) allow to determine their molecular brightness. **b** Conversion of pixel intensities associated with the plasma membrane (PM) into particle numbers based on the molecular brightness of the labels for ligands (L) and receptors (R). **c** Titrations of trisNTA-Alexa647 and **d** IL-4-ATTO647N on cells expressing IL-4Rα*. Fitting a model function for 1:1 binding reproduces published affinities of trisNTA ($34 \pm 8$ nM) and IL-4 ($0.14 \pm 0.04$ nM) towards IL-4Rα*[28,39]. Concentrations of free ligand (x-axis) stem from FCS measurements in the supernatant. Markers and error bars represent mean ± SD of 13–155 cells (y-axis, precise number of cells for each concentration provided in the Source Data file) or 18 FCS runs (x-axis). Experiments were performed once (**c**) or twice (**d**). **e** Varying average occupancy of BCNK bearing receptors at the plasma membrane at fixed

concentrations of IL-4-ATTO647N (color code according to Fig. 1a). Bars and error bars represent mean ± SD of single-cell measurements pooled from one (Lyn-eGFP, T18B), three (IL-4Rα*) or two (else) independent experiments. Descriptive statistics containing the number of cells measured for each receptor mutant are listed in Supplementary Table 3. **f** Gaussian-shaped probability distributions of single-cell averages of receptor occupancy (averages denoted in the panels). IL-4Rα* and Lyn-eGFP (gray) show stable means at both concentration levels ($K_d$ of natural ligand 0.15 nM[28]), whereas for non-saturated receptor mutants (lower affinity) the occupancy increases with higher IL-4 concentrations. Bimodal Gaussian distributions in or close to the activation loop (blue) indicate two co-existing receptor states with different affinity. Descriptive statistics of all mutants at both concentrations are listed in Supplementary Table 3 and Supplementary Table 4. Source data are provided as a Source Data file.

IL-4Rα* and $0.19 \pm 0.05$ μm²s⁻¹ for the receptor mutant K97B (single-letter code specifying the location (K97) of BCNK (B) replacement; numbering according to the native gene) were in agreement with previous measurements[30,38] (Supplementary Fig. 5). Cross-correlation amplitudes rise with co-diffusion of differently colored labels coupled to the same protein (Fig. 1d). The upper limit was marked by IL-4-ATTO647N binding to eGFP-tagged IL-4Rα*[34], whereas cross-correlation was close to zero for probing a non-interacting fluorescent membrane marker (Lyn-eGFP) (Fig. 1e). Eliminating a hotspot for ligand binding by BCNK incorporation (Y13B) abolished cross-correlation between receptors and ligand, whereas BCNK at a non-neutralizing position (K97B) produced significant cross-correlation of almost the same levels as IL-4Rα*. Thus, ligand binding capability of mutant K97B was preserved. Analysis of click-labeled receptor mutant (K97B-Cy5) in the absence of ligand also showed significant cross-correlation confirming co-diffusion between conjugated Cy5-tet2 and the eGFP-tagged receptors. However, the amplitude reached $38\% \pm 3\%$ of IL-4Rα* indicating the existence of non-labeled species at the cell

surface. Confirming the high degree of specificity of iEDDAC, labeling cells expressing the membrane marker with tetrazine showed no cross-correlation. Thus, covalent click labeling with the BCNK/tetrazine system was successful, albeit incomplete in terms of reaction yield.

## Cellular quantification by brightness-calibrated ratiometric imaging (BCRI)

Intracellular FCCS measurements are elaborate. To extract molecular information directly from confocal images, we developed brightness-calibrated ratiometric imaging (BCRI). The workflow combines single-point FCS for molecular brightness measurements of the used labels (eGFP, trisNTA-, IL-4-, and tetrazine-conjugated dyes) with confocal imaging under controlled instrumental settings, such that pixel intensities can be converted into particle numbers and local concentrations (Fig. 2a and Supplementary Fig. 6; for extended description see Supplementary Information). To generate reliable concentration data by image calibration, it is important to determine the exact photon yield reflected by the counts per particle (CPP) for a

given fluorescence detection setting. Therefore, the CPP was assessed by FCS measurements both in cells and in solution (Supplementary Fig. 7). Solution measurements can then be used to cover the typical range of excitation power densities applied for imaging (Supplementary Fig. 8). Once these factors are established, the day-to-day BCRI routine only requires monitoring of instrumental instabilities as the total laser power (Supplementary Fig. 9) and the optical performance of the microscope (Supplementary Fig. 10). To evaluate the cell surface as a reaction compartment, we employed a threshold-based segmentation algorithm (Fig. 2b, yellow mask). As a result, absolute and ratiometric numbers, like surface concentrations and the fraction of labeled receptors, can be determined from confocal images of individual cells.

To test BCRI with well-defined biochemical systems, we first quantified successive binding of trisNTA-Alexa647 to the His-tag at the N-terminus of the receptor[30,39]. The CPPs were measured with FCS in solution and showed linear dependence on laser power for both eGFP and trisNTA-Alexa647 (Supplementary Fig. 11). Considering non-specific binding of trisNTA-Alexa647 at the cell surface (Supplementary Fig. 12), saturated the ligand/receptor ratio at 100% occupancy (Fig. 2c). Thus, BCRI confirmed the 1:1 stoichiometry for this interaction and showed that the entire population of IL-4Rα* is accessible. Second, we titrated the fluorescently labeled natural ligand IL-4-ATTO647N. Taking into account that 32% of the sample were non-labeled (Supplementary Fig. 13), the titration curve saturated the ligand/receptor ratio at 100% occupancy as well (Fig. 2d). The 1:1 ratio of IL-4 binding is well established[26,27]. We like to note that reproducing the expected saturation levels for a given stoichiometry provide means to validate the correct CPPs used for image calibration via BCRI.

To push cellular quantification even further, we determined the concentration of ligands in the supernatant by single-color FCS. Since only active binding partners at the plasma membrane participate in the equilibrium, it is necessary to account for compound losses during titration. For a given concentration of free ligand, the fraction of bound receptors as determined by BCRI can be approximated with the Hill equation (Supplementary Equation 8). The dissociation constants for trisNTA ($34 \pm 8$ nM) and IL-4 ($0.14 \pm 0.04$ nM) were in excellent agreement with literature[28,39], highlighting the power of BCRI for bridging cellular image quantification with solution biochemistry.

## Functional consequences of BCNK incorporation

Encouraged by the consistency of these results, we turned to GCE-expressed receptor mutants bearing BCNK in their extracellular domain (Fig. 1a). Applying IL-4-ATTO647N in excess produced cell populations for which receptor occupancy was independent of the expression level of the particular cell (Supplementary Fig. 14). The population averages outlined the well-known energy landscape for binding (Fig. 2e and Supplementary Table 3): strong inhibition was observed for BCNK located in neutralizing positions at the center of the so-called binding clusters I and II (Y13B, D72B, and Y183B), which were previously identified as enthalpic hot-spots[40]. IL-4 binding was low but still above that of the plasma membrane marker Lyn-eGFP and mirrored the affinity ranking that was reported for corresponding alanine mutants (Y13A>D72A>Y183A)[29]. In contrast, ligand binding was significant but reduced as compared to IL-4Rα* in all other locations, most severely within the activation loop, a conformational sensitive region with respect to ligand-occupancy that was previously suggested to be crucial for activation of Janus kinases at the cytoplasmic tail of the receptor[41].

To address binding behavior in more detail, we considered a second concentration of IL-4-ATTO647N. As expected, above $K_d$, the occupancies measured for IL-4Rα* and the membrane marker were stable. However, with the exception of T18B, increasing IL-4-ATTO647N concentrations shifted the Gaussian-shaped cellular distributions towards higher occupancy (Fig. 2f and Supplementary

Fig. 15 and Supplementary Table 4). This elastic response suggests that the BCNK-bearing receptors suffer from reduced affinity towards IL-4, potentially reflecting partially steric hindrance in combination with subtle but well-defined structural changes. In contrast, a stable fraction of T18B receptor mutants could not be rescued by increased ligand concentration, suggesting this subpopulation to be trapped in an inactive state. Interestingly, two conformationally distinct populations were observed when BCNK locates in or close to the activation loop. For both receptor mutants (E141B and E189B), we deduce bimodal Gaussian distributions representing conformational substates with distinct ligand binding affinity. For the mutant E141B, with increased ligand concentration, the relative contributions of the substates were roughly maintained. In contrast, when BCNK locates in position X of the highly conserved WSXWS motif (E189B), a shifted high-affinity state is populated at the expense of the low affinity state. Thus, the activation loop indeed features a switch-like behavior with respect to ligand occupancy[41]. Taken together, BCRI showed that BCNK incorporation at different locations produced considerable functional heterogeneity.

## Conditions for click labeling at the cell surface

To be able to characterize click-labeled receptors, imaging was extended to three color channels such that we could probe eGFP-tagged receptors simultaneously for IL-4-ATTO647N binding and Alexa568-tet1 conjugation. Single-color controls addressed potential artefacts associated with cross-excitation and -detection (Supplementary Fig. 16). The fraction of click-labeled receptors, referred to as click efficiency (CE), was quantified based on Alexa568 signal referenced by the total pool of eGFP-tagged receptors. Click labeling receptor mutant K97B under the same conditions as used for the FCCS assay, showed a cellular average of CE = $20\% \pm 4\%$ (Fig. 3a). Thus, in agreement with reduced cross-correlation amplitudes, BCRI revealed a significant fraction of non-labeled receptors in the click channel. Labeling control cells expressing receptors in the absence of BCNK resulted in background levels of about one receptor per $\mu m^2$, demonstrating the impressive chemo-selectivity of the tetrazine-BCNK pair (Supplementary Fig. 17).

Calibration of the eGFP-channels showed that the cellular distribution of expression levels was discontinuous: a smaller sub-population of cells expressed less than ~200 receptors per $\mu m^2$, while most cells were showing a large spread (Fig. 3a). For receptor mutant K97B, covalent labeling and ligand binding were mutually independent in contrast to the mutant E141B where tetrazine conjugation impaired ligand binding (Supplementary Fig. 18). Thus, BCRI reveals functional side effects of site-specific click-labeling. Importantly, all tested receptor mutants showed average values of CE that were independent of cellular expression levels. In fact, repeated CE-measurements of mutant K97B by imaging of 15–40 hand-picked cells steadily reproduced CE = $19\% \pm 3\%$, which was entirely consistent with blindly processed images comprising hundreds of cells within a larger field of view (FOV, 213 $\mu m$ x 213 $\mu m$) (Supplementary Fig. 19). Thus, although click labeling was incomplete, CE determined by BCRI represents a remarkably robust readout related to the target protein, which is crucial for any subsequent biochemical analysis.

To clarify mechanistic aspects of live-cell labeling, we systematically varied the labeling conditions. Click-labeling cells expressing the mutant K97B with increasing concentrations of Alexa568-tet1 showed a distinct sublinear increase of CE (Fig. 3b). Instead of a linear dependence expected for pseudo-first-order conditions, the curve resembles a saturating two-state behavior most likely referring to the spatial heterogeneity of the system, which encompasses bulk solution and a thin surface layer containing extracellular protein domains and glycocalyx around the cells. In line with this notion, the kinetics showed a pronounced biphasic behavior: a rapid initial rise followed by a shallow increase (Fig. 3c). Fitting a double-exponential function

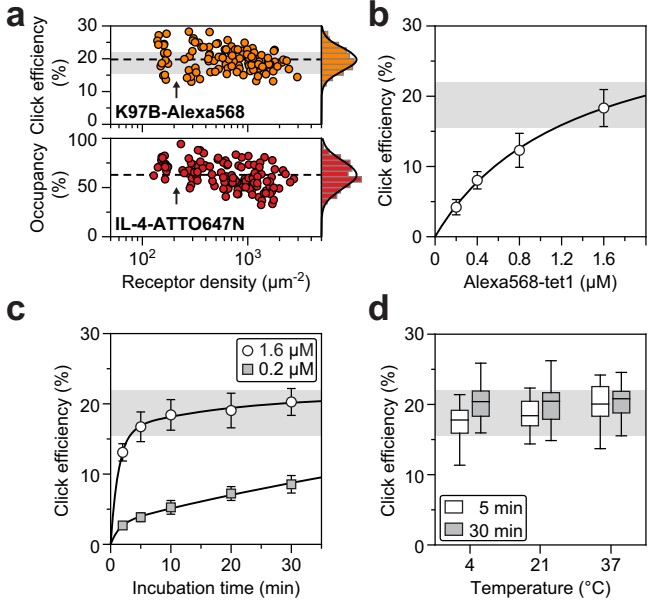

**Fig. 3 | Parameter space for covalent tetrazine labeling of receptor mutant K97B at the cell surface. a** Single-cell average of click efficiency (CE) and receptor occupancy as a function of single-cell expression levels based on eGFP. The CE of K97B measured under standard labeling conditions is indicated by the gray shading (19% ± 3%, mean ± SD). The range of observed expression levels is discontinuous (arrows). The average CE and occupancy in this experiment (*n* = 119 cells) is indicated by the dashed lines. **b** CE as a function of Alexa568-tet1 concentration (5 min incubation time); fitting follows a sublinear two-state saturation function. Markers and error bars represent mean ± SD of 20 cells. **c** CE as a function of incubation time (0.2 and 1.6 μM Alexa568-tet1); fitting follows a double exponential function. Markers and error bars represent mean ± SD of 50 (5 min) and 20 cells (else) in case of 0.2 μM or 20 (2 min) and 40 cells (else) in case of 1.6 μM Alexa568-tet1. **d** Influence of temperature on CE (1.6 μM Alexa568-tet1) after 5 and 30 min incubation time. Box-and-whisker plots indicate first and third quartile (box), median (horizontal line), and 1.5 times the interquartile range (whiskers) of 40 (30 min, 4 °C) or 20 cells (else). All experiments were performed once. Source data are provided as a Source Data file.

revealed that the fraction of CE during the initial phase dropped from 16% (1.6 μM) to 3% (0.2 μM) suggesting that the first 1–2 min dictate the overall yield. Estimating the volume of tetrazine corresponding to total amount of cell-bound receptors (Fig. 3a; 150 receptors per μm²), results in a shell thickness between 1.5–0.2 μm for bulk concentrations between 0.2–1.6 μM, respectively. These dimensions suggest that the overall yield may be sensitive to different fractions of tetrazine residing initially inside the glycocalyx layer (<100 nm thickness)[42]. As indicated by the sublinear concentration-dependence, excluded volume effects may lead to reduced local concentrations in the vicinity of the receptors. In agreement with this interpretation, the time constants for initial coupling depend weakly on bulk concentration (0.68 min⁻¹ for 0.2 μM and 0.77 min⁻¹ for 1.6 μM) and product formation appears to be dominated by the intrinsic reaction rate. In contrast, the second phase is dominated by diffusive exchange showing a distinct concentration-dependence (0.01 min⁻¹ for 0.2 μM and 0.05 min⁻¹ for 1.6 μM) of the same order of magnitude that was reported in methanol/water mixtures[9]. Thus, negatively charged Alexa568-tet1 must be replenished through a negatively charged polysaccharide layer containing glycosaminoglycan[43], which apparently slows down the click reaction at the cell surface by more than ten-fold.

Concentrations above 1.6 μM started to show abnormal cellular phenotypes, therefore we chose this concentration as an upper limit. To minimize fluid phase uptake of non-conjugated dye, routine labeling was performed with pre-chilled cells on ice. However, to explicitly test temperature dependence, we compared CEs for

identical labeling conditions at 4 °C, 22 °C, and 37 °C. The temperature shifts had surprisingly small effects, even at 37 °C where cellular receptor trafficking is released. (Fig. 3d). At short labeling times a shallow increase may be related to thermally accelerated reaction rates. In summary, these results establish important features of this system, robust labeling conditions in conjunction with a stable fraction of surface-accessible receptors at the plasma membrane during manipulation and detection of the cells.

## Site-dependent variations in click efficiency

Having established BCRI and a stable cell labeling procedure, we aimed to elucidate physico-chemical properties of the target protein. Screening all 14 receptor mutants under identical labeling conditions revealed reproducible site-specific variations of CE (Fig. 4a). In addition, we measured all 11 non-neutralizing receptor mutants after pre-incubation with saturating concentrations of IL-4. Under both conditions, the N-terminal domain D1 exhibited on average a higher CE than D2, possibly reflecting lower accessibility in membrane-proximal regions. With few exceptions, T18B (no change in CE), S44B and E189B (gain in CE), the site-specific pattern of non-neutralizing BCNK incorporation sites was largely reproduced when click-labeling occupied receptors, albeit on a significantly reduced level (compare gray to white boxes, Fig. 4a).

Protein-protein interactions following the induced fit mechanism can freeze backbone motions[44]. We therefore asked whether such a mechanism may explain the overall reduced reaction yield in the occupied state. To test this, we performed molecular dynamics (MD) simulations of the extracellular IL-4Rα domains with or without IL-4 ligand (Supplementary Fig. 20) starting from published crystal structures[27]. Computed B-factors from atomic fluctuations of the peptide bond at potential BCNK incorporation sites showed larger magnitudes as those extracted from experimental B-factors[45], but reproduced fairly well the crystallographic pattern (Fig. 4b and Supplementary Fig. 21)[26,27]. MD-derived amplitudes were indeed decreased in the IL-4-occupied state. This encompassed all D1-sites, but also Tyr183 and Glu189 located in the activation loop of D2, which connect mechanically to the IL-4 binding epitope. The data suggest that backbone motions play a supportive role for the overall reaction yield. However, neither MD-derived nor experimental mobility patterns showed a significant correspondence to site-specific variations of CE.

To analyze the chemical environment of the reactive BCNK ring in more specific detail, MD-analysis was extended to a subset of receptor mutants bearing BCNK at locations in D1. A correlation with the measured CE was not evident from steric considerations (Supplementary Movies 1–9). For example, in MD simulations of the receptor mutants K97B and E189B, representing minimum and maximum CE of the entire set, the terminal BCNK rings adopt almost identical positions within their respective average structures (Supplementary Fig. 22). In search for quantitative parameters, we designed a 'catchbox' of approximately tet1 dimensions and screened for atoms on the receptor surface that would fall into it along the simulation trajectory (Fig. 4c and Supplementary Figs. 23 and 24). For T18B and K22B, the distribution of distances from which atoms could enter the BCNK catchbox was exceptionally large and represented the dimension of the tetrazine-BCNK assembly (20–30 Å; Supplementary Fig. 25), while most other sites were limited to distances of about 10 Å (Fig. 4d and Supplementary Fig. 26). In contrast, some receptor mutants showed a confined contact area. In the receptor mutant E94B, having the same CE as broadly distributed K22B, the terminal ring of BCNK is buried in a hydrophobic pocket formed by Gly92 and Leu93 of the ligand (Fig. 4d). Statistical attempts to correlate CEs with certain classes of surrounding atoms or side chains for the entire data set were unsuccessful (Supplementary Table 6). In addition, we quantified the flexibility of the BCNK's terminal ring from MD trajectories. In agreement with the larger contact surface of T18B and K22B, the rotational flexibility of the terminal ring was

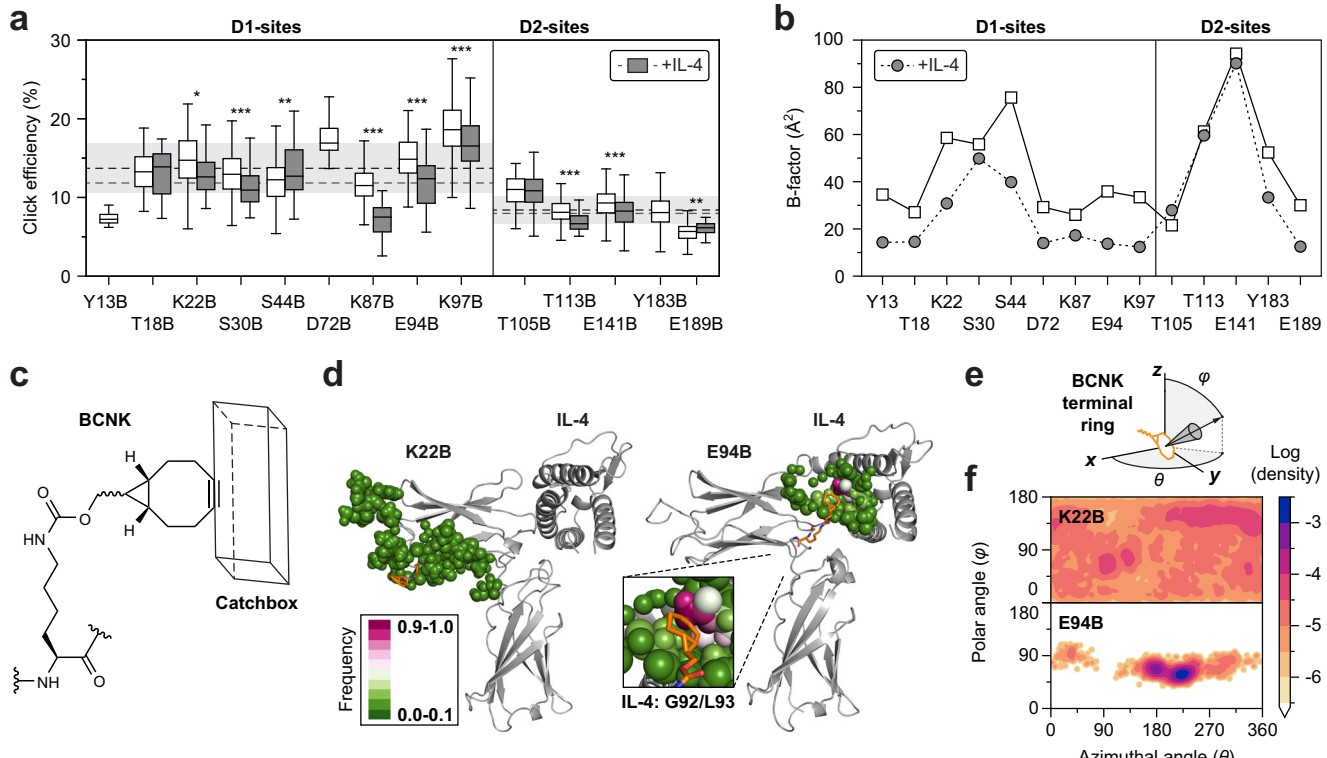

**Fig. 4 | Click labeling of BCNK at different locations on the protein surface.**
**a** Site-specific signatures of click efficiency (CE) in the absence and presence of IL-4 measured under identical labeling conditions. Domain-specific averages (neutralizing sites omitted) are slightly reduced in the presence of IL-4 (dashed lines). The range of two standard deviations (shaded, light gray) was calculated for each domain separately. Significance was tested with two-sided Welch's *t*-test. Asterisks indicate $p < 0.05$ (*; K22B: $p = 0.019$), $p < 0.01$ (**; S44B: $p = 0.0070$, E189B: $p = 0.0095$), $p < 0.001$ (***; S30B: $p = 0.0006$, K87B: $p < 0.0001$, K94B: $p < 0.0001$, K97B: $p < 0.0001$, T113B: $p = 0.0002$, E141B: $p < 0.0001$). Box-and-whisker plots indicate first and third quartile (box), median (horizontal line), and 1.5 times the interquartile range (whiskers). Descriptive statistics containing the number of cells measured for each condition and the number of biological replicates are listed in Supplementary Table 5. **b** B-factors of atoms forming the BNCK peptide bonds from

MD simulation of IL-4Rα (PDB: 3BPN) show reduced motion at key positions of IL-4 ligand binding. **c** Scheme of the terminal catchbox with dimensions resembling the tetrazine moiety (13.5 Å x 4.5 Å x 3.5 Å) used for sampling atoms during MD simulations. **d** Average structures from 250 ns MD trajectories of receptor mutants (K22B, E94B). Atoms entering the catchbox along the trajectory are highlighted ('spheres' representation, color code indicates the frequency of occurrence). Despite similar CE, the contact area for K22B is widely distributed, whereas in E94B the terminal ring populates a specific hydrophobic hotspot (zoom-in). **e** Definition of polar and azimuthal angles of the terminal BCNK ring as a measure for rotor flexibility. **f** Probability distribution of polar and azimuthal angles of the BCNK ring for the receptor mutants (K22B, E94B) corresponding to (**d**). Source data are provided as a Source Data file.

almost unimpaired, whereas in E94B, it is confined to particular polar and azimuthal angles (Fig. 4e, f and Supplementary Fig. 27). Thus, rotational ring flexibility can be ruled out as a crucial factor for CE as well. In conclusion, although chemical contacts and mobility of the reactive BCNK ring were remarkably diverse, none of the tested parameters could explain the local variations of click labeling.

Since CE could not be linked to properties of the target protein alone, we took the surrounding solvent into consideration. Protein solvation is believed to have a strong impact on conformation and dynamics, in particular for membrane-embedded proteins[46]. Solvation effects were computed by solving the Poisson-Boltzmann equation in the framework of the polarizable continuum model[47–49]. MD average structures of the receptor mutants formed the solutes of single point solvation free energy (SFE) calculations considering vacuum-to-water transfer. Here, the net solvation-free energy is a sum of polar and non-polar terms where the decisive contribution is made by the polarization of the solvent (see sketched field lines and residual polarization charges at the solute-solvent boundary in Fig. 5a and compare partial contributions in Supplementary Table 7). Interestingly, an intriguing correlation was observed between bulk polarization and experimentally determined CE values (Fig. 5b). Although the physicochemical driving force for the reaction rate remains elusive, this correlation suggests that the iEDDAC reaction is driven by a mechanism that is

critically conditioned by the surrounding solvent. The average position of the reactive BCNK moiety can be considered to be both productive and representative for the reaction yield. However, as the position of the charged fluorescent tetrazine as a reaction partner is neglected in the observed correlation, a mechanism that involves charge-mediated docking of reactants prior to conjugation is not supported.

## Ligand-dependent variations in click efficiency
The similarity of site-specific signatures when click-labeling IL-4Rα in the presence or absence of IL-4 suggested that CE is somehow linked to structural similarities of both receptor states. Indeed, bulk polarization depends globally on the sum of energy contributions of the entire protein domain, where BCNK is embedded, and hence, conformations. We, therefore, asked whether a correlative link between CE signatures and protein structure could be revealed. Analyzing the differential, IL-4-dependent change in click efficiency (differential CE; CE of occupied receptors *vs.* non-occupied receptors for the same receptor mutant) for all non-neutralizing sites shows small but distinct deviations from the pattern that were significant themselves. Similar to absolute CE, the magnitude of differential CE was domain-specific (D1>D2; Fig. 4a).

To assess structural differences displayed in IL-4Rα crystal structures including or devoid of IL-4, we superimposed the IL-4Rα unit of

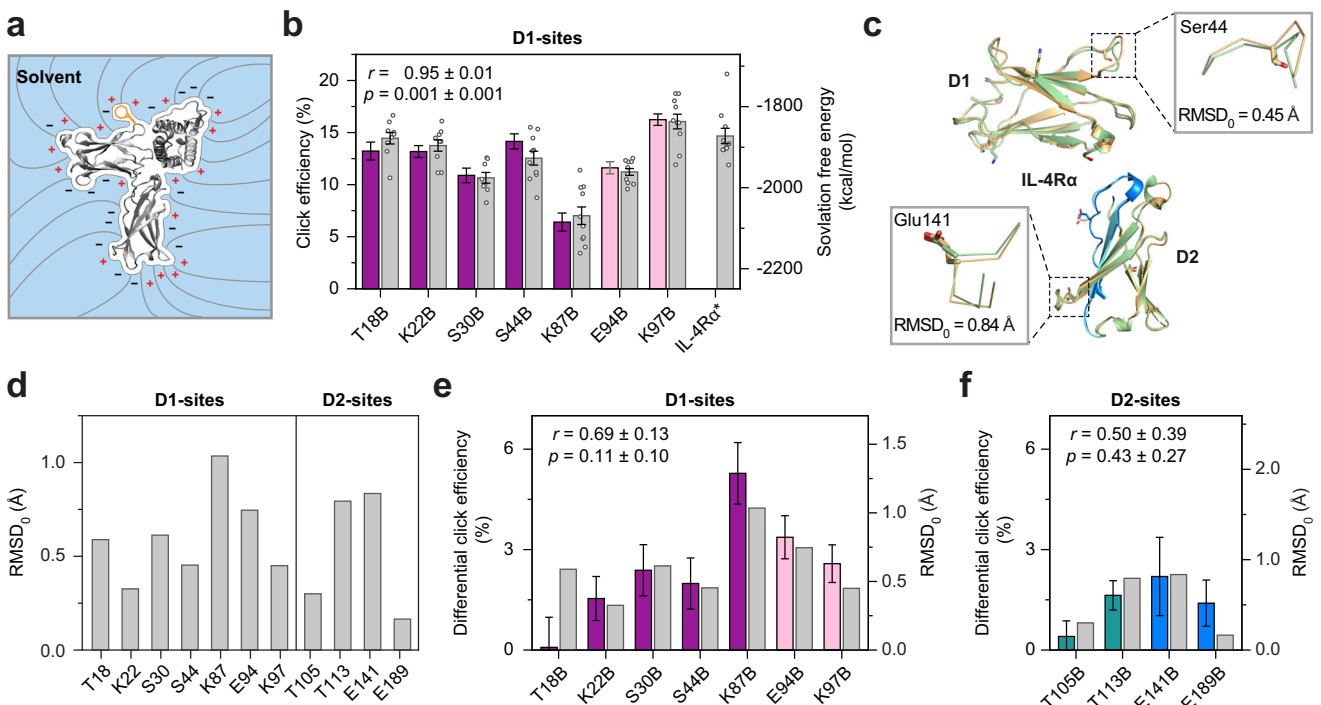

**Fig. 5 | Structural basis for site-specific variations of click efficiency. a** Schematic representation of partial contributions to the solvation free energy (SFE): cavitation (creating a void, white), dispersion (attractive van der Waals interactions at the solute-solvent interface) and polarization (sketched field lines and residual polarization charges in blue and red). **b** Strong correlation (Pearson correlation coefficient $r = 0.95$) between click efficiency (CE, colored bars according to Fig. 1a) and SFE (gray bars) for each of the average structures including BCNK at indicated positions. Bars and error bars represent mean ± SEM of single-cell measurements (CE, see Supplementary Table 5 for precise $n$ values) or of eight to ten average structures derived from sub-trajectories (SFE). **c** Structural alignment of the ectodomains of IL-4Rα in a complex with IL-13 (palegreen; PDB: 3BPO) or IL-4 (wheat; PDB: 3BPN). The flexible activation loop, connecting the transmembrane domain with the IL-4 binding epitope, shows minimal deviations (green cyan and marine blue). Backbone shifts of exemplary residues in the D1 (Ser44) and D2 (Glu141) domain upon IL-4 binding are highlighted (insets: backbone Cα-atoms in 'stick' representation; $RMSD_0$ represents the translocation of the BCNK junction without neighboring Cα-atoms). **d** RMSD of the BCNK junction at different incorporation sites. **e, f** Strong correlation ($r = 0.69$ for D1 and $r = 0.50$ for D2 domain) of backbone variations (gray bars) with absolute values of differential CE for the N-terminal D1 and membrane-proximal D2 domain. Bars and error bars of differential CE are defined as mean ± SEM of CE differences in the presence and absence of IL-4 (see Supplementary Table 5 for precise $n$ values). Error was estimated by uncertainty propagation assuming independent variables. Source data are provided as a Source Data file.

ternary type 2 complexes[27], in which either IL-4 or IL-13 is complexed by the same receptor dimer (Fig. 5c). Both ternary complexes are distinguished by their binding sequence: while IL-4 is initially engaged by IL-4Rα, IL-13 recognizes the second receptor subunit IL-13Rα1. Since the recruited second receptor subunit provides very small enthalpic contributions for the ligand[27,30], the complex with IL-13 served as a proxy for non-occupied IL-4Rα, for which no crystal structure was available. Evaluating the structural alignment (Fig. 5d) at the different incorporation sites showed that the ligand-dependent displacement of BCNK sites varied between 0.165 Å (Glu189) and 1.035 Å (Lys87). Apart from Ser44, which is located in a strongly bent loop close to the IL-4 binding epitope, RMSD values did not undergo significant changes when including more neighboring Cα-atoms (Supplementary Fig. 28 and Supplementary Table 8).

Considering each domain separately, the RMSD pattern correlates surprisingly well with the measured absolute values of differential CE (Fig. 5e, f and Supplementary Fig. 29). Thus, the shifts in CE apparently reflect conformational changes in the IL-4Rα structure incurred upon IL-4 ligand binding. The different linear conversion factors, which scale structural translocation (RMSD) to the measured change in reaction yield (differential CE) for each domain, reflect most likely the topology of the receptor with respect to the membrane plane. Within D1, the correlation failed for T18B with a stable fraction of inactive conformations (*cf.* Supplementary Fig. 14). Within D2, the correlation failed for E189B, the pivotal amino acid of a membrane-dependent conformational switch, which is not apparent in crystal structures.

Intriguingly, the behavior of the outlier E189B agrees qualitatively with a model according to which the glutamate at position X of the WSXWS-motif gains accessibility in receptor orientations representing the occupied on-state[41].

## Discussion

The chemical diversity of peptidogenic building blocks of proteins calls for technologies that resolve functional states at the resolution of single amino acids. Accordingly, mutational studies on proteins in tissues and cells have mounted a wealth of functional information. With increasing coverage of structural biology data, many of these protein functions are now attributed to detailed molecular models striving for mechanistic understanding at atomic resolution. However, to date, the endeavor of cross-validating those mechanistic models based on solution structures within the context of living cells still represents an extremely ambitious task. This knowledge gap is particularly grave in cell signaling[50], where signals encoded as conformational substates within complexes of different compositions are trafficking through a very heterogeneous environment as comprised by the different cellular compartments. We here demonstrate that bioorthogonal click labeling in conjunction with quantitative microscopy provide a promising means to approach this issue.

We develop BCRI as an instructive yet frugal microscopic approach to quantify fluorescent receptor subpopulations in living cells by confocal microscopy. The experimental protocol achieves two goals: first, pixel intensities convert into numbers of molecules and,

subsequently, local concentrations. Second, calibration is extended to multiple color channels from where molecular ratios can be computed. The molecular ratio represents a substantial biochemical observable as it reflects functional subpopulations of target proteins which can be referenced against each other. Once the molecular brightness of the fluorescent labels is determined, BCRI could achieve an about ten-fold larger throughput as compared to intracellular FCS. Thus, BCRI constitutes a promising asset for fine-tuning bioorthogonal labeling ventures in both divisions, the chemical toolbox as well as cell biological implementation.

To showcase the abilities of the BCRI approach, we first address functional consequences of the insertion of BCNK at different positions in the extracellular domain of IL-4Rα with a single-cell binding assay. As expected, IL-4 binding is mostly abolished when the bulky BCNK locates within the binding epitope, whereas it remains preserved for all non-neutralizing sites. However, quantifications show that BCNK reduces ligand affinity in a site-specific manner. In particular, sites in or close to the activation loop show bimodal distributions of reduced affinity, which hints at strong allosteric effects. Susceptibility towards receptor occupancy in combination with structural vulnerability are a hallmark of conformational changes in signal transduction. It appears quite remarkable that site-specific bioorthogonal labeling in combination with cellular quantifications enable to pinpoint such features within the receptor structure. IL-4 binding assays in the presence of labeled BCNK produced similar results, suggesting that the BCNK incorporation represents the dominant perturbation whereas additional click labeling has minor functional consequences.

Since methods for cellular quantifications were limited, effects of varying click labeling conditions at the cell surface have not yet been systematically addressed. Indeed, for iEDDAC employing the BCNK-tetrazine pair, we observe a much lower degree of labeling for cell-bound receptors as compared to free solution[9]. Tetrazine association follows a comparatively slow dual phase kinetics for which saturation is never reached. In addition, the reaction rates show a sublinear concentration dependence. Thus, click labeling at the cell surface cannot be modelled by diffusion-limited pseudo-first order kinetics. It seems the complicated cell topology comprised by a ruffled plasma membrane, a charged glycocalyx layer and the heterogeneous spatial distribution of cell-bound receptors, require a more sophisticated description of the diffusion part. However, the click reaction still remains pseudo-first order in the sense that local receptor-bound BCNK concentrations are low enough to have negligible effect on the reaction rate. This thermodynamic feature allows determination of a representative cell-average by measuring a relatively small number of hand-selected cells that show suitable signal levels for confocal imaging.

Environmental parameters for iEEDAC-mediated conjugation like tetrazine concentration, incubation time, and temperature were mainly limited by abnormal cellular phenotypes and our optimized standard labeling conditions are in agreement with reported protocols[9,13,18,19]. Importantly, the fraction of click-labeled receptors measured for one and the same receptor mutant in a series of experimental repetitions using standardized conditions revealed a high degree of reproducibility. While the surface density of the GCE-expressed, BCNK-bearing receptors varied over three orders of magnitude, the CE measured under identical conditions remained stable within a few percent of error. Thus, above the background of instrumental (BCRI) and procedural (cell preparations) noise, site-specific variations of CE become meaningful.

Validation of BCNK incorporation sites under standardized conditions produced a specific signature of varying click efficiencies similar to observations made with G-protein coupled receptors[19]. We attempted to search for correlative patterns in the crystal structure into which BCNK was modeled. Similar to CE, many parameters deduced from MD simulations showed significant site-specific

variations. Although it stands to reason that stereochemical constraints may influence the success rate for iEDDAC, neither flexibility of the BCNK as a whole nor the terminal ring showed correlations with the corresponding CE. Likewise, sampling the chemical environment in the vicinity of the reactive ring failed to provide any correlative relationships. Instead, we found that bulk polarization of the water-embedded target protein, calculated for average structures of the different receptor mutants bearing BCNK at different locations, strongly mirrors the experimental click efficiencies. Thus, in absolute terms, it appears that the better a particular receptor structure was stabilized by the solvent (more negative SFE), the less reactive its BCNK moiety turned out to be. Assuming comparable levels of mean polarization per solute charge in the receptor mutants, the up/down-shifts in SFE should stem from altered electrostatic potentials on the solute-solvent boundary. Due to the long-range nature of electrostatic interactions, it is perfectly conceivable that the insertion of BCNK within a cluster of polar or charged amino acids can lead to a significant perturbation of a great many neutralizing relations, hence causing changes in polarization.

Encouraged by the intriguing link between structural topology of the BCNK bearing receptor mutants and absolute CE, we investigated the effect of ligand-induced structural changes. Comparing CE signatures of free and IL-4-bound receptors shows small but distinct shifts, which correlate remarkably well with structural deformations in the extracellular domain of IL-4Rα. The sub-angstrom scale of backbone translocations is strikingly small, raising the question of how such a great sensitivity is conceivable. We suppose that BCNK- and IL-4-induced structural changes are largely uncoupled. According to this assumption, structural perturbations induced by BCNK are independent of receptor occupancy and site-dependent contributions mostly cancel when computing the differential CE. Since IL-4 imposes

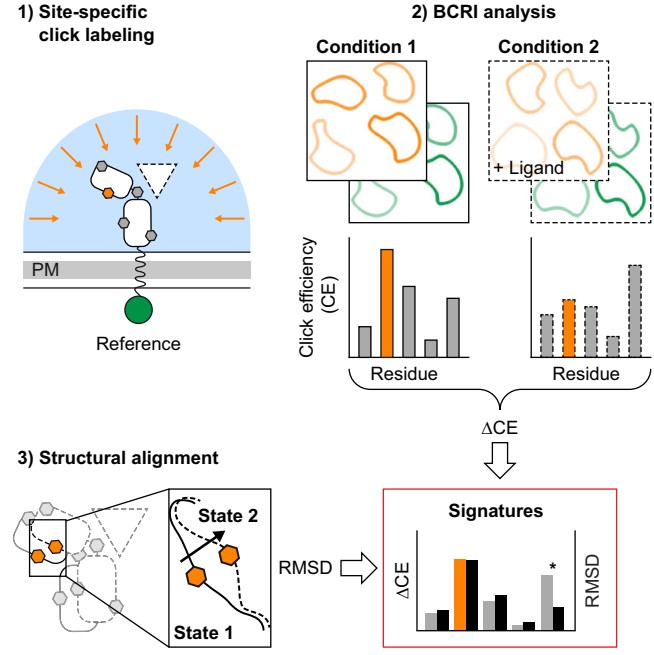

**Fig. 6 | Click labeling and BCRI detect sub-angstrom shifts of receptor conformations in their native environment.** Click labeling in combination with BCRI provides protein structure-related patterns of a site-specific reaction yield (click efficiency; CE). The local changes of a protein's click efficiency measured under varying conditions (e.g. of a receptor in its occupied and non-occupied state) reflect conformational shifts of the protein backbone down to the sub-angstrom range. By comparing the magnitude of shifted click efficiencies (ΔCE) with signatures of backbone displacements from structural alignments (RMSD), in-solution crystallographic data can be cross-validated in living cells. Regions of increased heterogeneity caused by additional cues like the plasma membrane or unknown ligands could be identified (asterisk).

structural changes of the protein backbone as well, the three-dimensional organization of charge centers is nonetheless affected. Thus, residual changes in CE seem to scale in linear approximation with small BCNK translocations within a quasi-static energy landscape.

The capability to sense conformational changes in the tertiary structure of a target protein in living cells is auspicious. These findings suggest a generic strategy to approach conformational substates at the cell surface (Fig. 6). CE signatures may be regarded as fingerprints that allow to associate protein conformations with certain experimental conditions and hence facilitate to identify molecular constraints that define these substates. Thus, in perspective, bioorthogonal labeling in combination with BCRI could shed new light on dynamic protein transitions involved in signal transduction at the plasma membrane and potentially other dynamic processes in molecular cell biology.

## Methods

### Chemicals
Ultrapure water was supplied by a purification system (Milli-Q, Merck). 1 mg lyophilized Cy-5-tet2 ester (Click Chemistry Tools) was dissolved in DMSO and stored at −80 °C. Prior to click labeling, tetrazine derivative DMSO stock solutions were diluted to working concentrations of 1.6 μM in air buffer. The soluble His-tag affinity dye trisNTA-Alexa647 was used as described[30]. BCNK was purchased from Synaffix.

### Recombinant IL-4-ATTO647N
Human recombinant IL-4 (Table 1) was produced and labeled at position N38C with maleimide-functionalized ATTO647N (ATTO-TEC) as described[37].

The dye-to-protein ratio was determined by absorption spectroscopy (JASCO photometer V-780) in 100 μl quartz cuvettes (105.201 QS; Hellma) at a scan speed of 400 nm min$^{-1}$. Absorption spectra of IL-4-ATTO647N and ATTO647N-COOH were measured in PBS (pH 7.4) in the 250–725 nm wavelength range with a step size of 1 nm and a spectral slit width of 1 nm. To estimate the degree of labeling (DOL), the dye (ATTO647N) and protein (IL-4) concentrations were calculated based on the relation.

$$DOL = \frac{c_{dye}}{c_{protein}} = \frac{A_{dye,max}\varepsilon_{prot,280}}{\left(A_{dye,280} - A_{dye,max}CF_{280}\right)\varepsilon_{dye,max}}. \quad (1)$$

where $A_{dye,280}$ and $A_{dye,max}$ is the absorbance of the dye-conjugate at 280 nm and at the absorption maximum, respectively, $\varepsilon_{dye,max}$ the extinction coefficient of the dye at the absorption maximum, $\varepsilon_{prot,280}$ the extinction coefficient of the protein at 280 nm, and $CF_{280}$ the relative absorbance of the dye at 280 nm. Inserting $A_{dye,max} = 0.448$, $A_{dye,280} = 0.0614$, $\varepsilon_{dye,max} = 150\,000\ \text{M}^{-1}\,\text{cm}^{-1}$, $\varepsilon_{prot,280} = 8855\ \text{M}^{-1}\,\text{cm}^{-1}$[51], and $CF_{280} = 0.05$ yields a DOL of 68% for IL-4-ATTO647N.

### Receptor constructs
Cloning and expression of the reference receptor IL-4Rα* (Table 2) in HEK293T cells (ATCC CRL-3216) was extensively described (pNHis-IL4-Ram266-eGFP-N2)[30,34,52]. The protein construct IL-4Rα* is composed of an exposed N-terminal hexahistidine stretch (His-tag), a truncated, signaling-deficient version of Interleukin-4 receptor alpha (isoform 1; Uniprot: P24394-1) fused with a C-terminal, cytoplasmic eGFP. A library of 14 receptor mutants with different TAG positions in the extracellular domain (pCMV_H6-IL4Ram266-X>B-eGFP; X = [Y19, T24, K28, S36, S50,

D78, K93, E100, K103, T111, E147, Y189, E195]) were derived based on IL-4Rα* by site-directed mutagenesis (GeneArt, Invitrogen). Note that, due to the His-tag, numbering is shifted by +6 amino acids with respect to mature wild-type. In addition, for a second construct of receptor mutant K97B the C-terminal eGFP was removed (pCMV_H6-IL4Ram266-K97B) for negative controls in FCCS and dual-channel BCRI. While the CMV-promotor was necessary for the GCE-mediated expression of receptor mutants (pCMV_H6-IL4Ram266-X>B-eGFP), microscopy required to express the non-modified control IL-4R* under the weaker SV40 promotor in early direction (pc2SV_NHis-IL4Ram266-EGFP-rc)[34]. The plasma membrane marker Lyn-eGFP (Table 2) was provided by Christian Bökel (University of Ulm, Germany).

### Cell culture and transfections
Adherent HEK293T cells were grown with DMEM supplemented with 10% FBS (Gibco) in T75 flasks (Thermo Fisher Scientific) under humidified atmosphere at 37 °C and 8.5% CO2. The cells were transfected 24 h post-seeding in 12-well multiwell plates (Thermo Fisher Scientific) at 40–60% confluence. The transfection reagent jetPRIME (Polyplus transfection) was used with a DNA-reagent ratio of [100 ng]: [0.3 μl], not exceeding in total 2 μg DNA per well. For TAG-codon suppression, plasmid DNA coding for the receptor mutants was transiently co-expressed with an orthogonal tRNA-synthetase/tRNA pair carrying the UAA[35]. The concomitant plasmid BCNK-RS[9,18] codes for the BCNK-synthetase driven by EF-1α promoter and, additionally, four copies of cognate tRNA driven each by a separate U6 promoter. Co-transfections were carried out with 0.3 μg receptor plasmid (pCMV_H6-IL4Ram266-X>B-eGFP) and 1.2 μg concomitant plasmid. To initiate GCE-mediated receptor expression, BCNK (stock 100 mM in 100 mM NaOH) was added 4 h after transfection to a final concentration of 0.5 mM, which was immediately neutralized with equal amounts of 100 mM HCl.

### Cell labeling
Manipulation of cells was performed in a home-made "air buffer" for life cell microscopy[30]: 150 mM NaCl, 20 mM HEPES pH 7.4, 20 mm D-(+)-trehalose, 15 mM glucose, 5.4 mM KCl, 0.85 mM MgSO$_4$, 0.6 mM CaCl$_2$, 0.15 mg ml$^{-1}$ bovine serum albumin (all components from Sigma Aldrich). For coating of glass slides, fibronectin (Roche Life Science) aliquots were prepared in Dulbecco's phosphate-buffered saline (Thermo Fisher Scientific) and stored at −20 °C.

For click labeling, 24 h post-transfection, adherent cells were washed with PBS (phosphate-buffered saline), harvested with 0.48 mM EDTA/PBS (Versene; Thermo Fisher Scientific), transferred to microcentrifuge tubes and pelleted at 180 x g on a cooled tabletop centrifuge (Eppendorf). The cells were resuspended in ice-cold air buffer, washed twice with ice-cold air buffer and stored for another 20 min on ice. During this phase 200 ng ml$^{-1}$ recombinant IL-4 (Gibco) could be added, when producing occupied receptors prior to labeling. Regular labeling in the absence or presence of IL-4 were performed with 1.6 μM fluorophore-tetrazine conjugates (tet1 and tet2) in air buffer for additional 5 min on ice. For seeding microscopy slides, the labeled cells were washed twice with ice-cold air buffer and transferred to another microcentrifuge tube (Eppendorf), washed carefully again three times with ice-cold air buffer and finally seeded in 8-well chambered glass-bottom slides (LabTek #1.5; Thermo Fisher Scientific). Glass-bottom slides have been pre-coated with 10 μg ml$^{-1}$ fibronectin for 1 h at 37 °C. Brightness-calibrated ratiometric imaging (BCRI) was performed within the next 3 h.

## Table 1 | Recombinant IL-4 protein used in this study

| Protein | Sequence | Notes |
|---|---|---|
| IL-4 | MHKCDITLQE IIKTLNSLTE QKTLCTELTV TDIFAASKCT TEKETFCRAA TVLRQFYSHH EKDTRCLGAT AQQFHRHKQL IRDLKRLDRN LWGLAGLNSC PVKEANQSTL ENFLERLKTI MREKYSKCSS | Initial methionine IL-4 (Uniprot: P05112; amino acids #25-#153) Substitutions N38C for labeling and F82D for folding |

**Table 2 | Membrane proteins IL-4Rα\* and Lyn-eGFP expressed in HEK293T**

| Protein | Sequence | Notes |
|---|---|---|
| IL-4Rα\* | MHHHHHHKVL QEPTCVSDYM SISTCEWKMN GPTNCSTELR LLYQLVFLLS EAHTCIPENN GGAGCVCHLL MDDVVSADNY TLDLWAGQQL LWKGSFKPSE HVKPRAPGNL TVHTNVSDTL LLTWSNPYPP DNYLYNHLTY AVNIWSENDP ADFRIYNVTY LEPSLRIAAS TLKSGISYRA RVRA-WAQCYN TTWSEWSPST KWHNSYREPF EQHLLLGVSV SCIVILAVCL LCYVSITKIK KEWWDQIPNP ARSRLVAIII QDAQGSQWEK RSADPPVVSK GEELFTGVVP ILVELDGDVN GHKFSVSGEG EGDA-TYGKLT LKFICTTGKL PVPWPTLVTT LTYGVQCFSR YPDHMKHDFF KSAMPEGYVQ ERTIFFKDDG NYKTRAEVKF EGDTLVNRIE LKGIDFKEDG NILGHKLEYN YNSHNVYIMA DKQKNGIKVN FKIRHNIEDG SVQLADHYQQ NTPIGDGPVL LPDNHYLSTQ SALSKDPNEK RDHMVLLEFV TAA-GITLGMD ELYK | His-tag: H6<br>IL-4Rα (Uniprot: P24394-1; amino acids #2-#266) with incor-poration sites X<br>Linker: ADPPVV<br>Fluorescent reporter: eGFP (Uniprot: P42212; amino acids #2-#238 with substitutions F64L, S65T) |
| Lyn-eGFP | MGCIKSKRKD NLNDDEAAMG CIKSKRKDNL NDDEAPVVSK GEELFTGVVP ILVELDGDVN GHKFSVSGEG EGDATYGKLT LKFICTTGKL PVPWPTLVTT LTYGVQCFSR YPDHMKQHDF FKSAMPEGYV QERTIFFKDD GNYKTRAEVK FEGDTLVNRI ELKGIDFKED GNILGHKLEY NYNSHNVYIM ADKQKNGIKV NFKIRHNIED GSVQLADHYQ QNTPIGDGPV LLPDNHYLST QSALSKDPNE KRDHMVLLEF VTAAGITLGM DELYK | Tandem of signals for acylation[72]<br>Linker: APVV<br>Fluorescent reporter: eGFP |

For quantification of GCE-mediated receptor expression, cells were prepared as for click labeling. After the 20 min incubation phase on ice, the plasma membrane was counterstained with 1 µg ml⁻¹ Cy5-NHS (GE Healthcare) dissolved in DPBS for 5 min on ice. Then the labeled cells were carefully washed three times with ice-cold air buffer and seeded on fibronectin-coated glass-bottom slides for calibrated imaging.

For ligand binding assays, transfected cells were seeded subconfluently in 8-well chambered glass-bottom slides (Thermo Fisher Scientific) as described above. To minimize non-specific adsorption to the glass surface, the glass slides were first pre-coated with 10 µg ml⁻¹ fibronectin and then 0.01% poly-L-lysine (Sigma-Aldrich) each for 1 h at 37 °C. After 1 h, when cells started to stretch on the support, IL-4-ATTO647N or trisNTA-Alexa647 was added at varying concentration to the supernatant (dilution series). Cells were incubated with ligands for another 30 min at room temperature prior to BCRI analysis.

### FCS and FCCS at the plasma membrane of living cells

Click-labeled or transfected cells were transferred to glass-bottom slides in air buffer as described above. To measure FCCS in the presence of IL-4 ligand, 20 nM IL-4 ATTO647N was added into the bulk and incubated for 30 min at room temperature.

For positioning the laser focus, transfected cells with similar fluorescence intensity levels in the green and red channel were selected showing homogeneous regions of fluorescence intensity at the bottom membrane in both color channels. Fluorescence fluctuations were recorded for 120 s, split into 6 runs à 20 s. Runs showing slow intensity drifts or intensity spikes were discarded. Due to unbound ligand in the supernatant, the autocorrelation functions of both channels were fitted with a model containing diffusion in 3D and 2D as well as a triplet component,

$$G(\tau) = \frac{1}{N} G_{\mathrm{T}}(\tau) G_{3\mathrm{D}}(\tau) G_{2\mathrm{D}}(\tau). \tag{2}$$

$$G_{\mathrm{T}}(\tau) = 1 + \frac{f_{\mathrm{T}}\, e^{-\frac{\tau}{\tau_{\mathrm{T}}}}}{1 - f_{\mathrm{T}}}. \tag{3}$$

$$G_{3\mathrm{D}}(t) = f_{3\mathrm{D}} \left(1 + \frac{\tau}{\tau_{3\mathrm{D}}}\right)^{-1} \left(1 + \frac{\tau}{\mathrm{AR}^2 \tau_{3\mathrm{D}}}\right)^{-\frac{1}{2}}. \tag{4}$$

$$G_{2\mathrm{D}}(t) = \frac{1 - f_{3\mathrm{D}}}{1 + \tau/\tau_{2\mathrm{D}}}. \tag{5}$$

where $N$ denotes the total particle number, $f_{\mathrm{T}}$ the fraction of particles in triplet state, $\tau_{\mathrm{T}}$ the characteristic triplet residence time, $\tau_{3\mathrm{D}}$ the diffusion time of freely diffusing particles, AR the axis ratio of the confocal observation volume, $\tau_{2\mathrm{D}}$ the diffusion time of membrane bound particles, and $f_{3\mathrm{D}}$ the fraction of molecules of the freely diffusing species ($n_{3\mathrm{D}} = f_{3\mathrm{D}} N$). For fitting, the axis ratio AR was fixed to 6, the triplet or blinking time was constrained to values between 1 and 100 µs, the 3D diffusion time to values between 100 µs and 3 ms, and the 2D diffusion time to values between 3 and 300 ms. Only data sets with at least 2 appropriate runs in both channels were considered for auto and cross-correlation analysis.

The molecular brightness or counts per particle (CPP) was calculated by the ratio of the average fluorescence intensity $F$ in the channel to the corresponding particle number $N$ (CPP = $F/N$). In FCCS, the cross-correlation function was fitted with a 2D diffusion model. Auto- and cross-correlation amplitudes were corrected for non-correlated background and 1% spectral cross-talk from the green into the red channel[53]. Diffusion coefficients were determined relative to freely diffusing calibration dyes ATTO488-COOH and ATTO655-COOH in water with known diffusion coefficients that were used to calibrate the observation volumes in each color channel (see section *BCRI−calibration of the confocal observation volume*).

### Confocal imaging

Confocal images were taken with a laser scanning microscope LSM780 (Carl Zeiss Microscopy), equipped with a ConfoCor3 unit, two avalanche photo-diode detectors (APDs) and a C-Apochromat 40x/1.2 water immersion objective (Carl Zeiss Microscopy). Images were recorded with the ZEN software (Carl Zeiss Microscopy) as 16-bit arrays in photon counting mode with common filter settings for eGFP (495−530 nm, band pass) and Alexa568-tet1 (580 nm, long pass) or IL-4-ATTO647N/trisNTA-Alexa647 (655 nm, long pass). The confocal pinhole was set to 37 µm (eGFP and Alexa568-tet imaging) or 40 µm (eGFP and IL-4-ATTO647N or trisNTA-Alexa647 imaging) and the pixel dwell time to values between 12−50 µs. Laser power was adjusted such that the fluorescence intensity of bright pixels was below 1 MHz to minimize saturation artifacts.

### Brightness-calibrated ratiometric imaging (BCRI)

The membrane of single-cell images was located with a custom *ImageJ* segmentation script (https://doi.org/10.17617/3.YDSJ2C, provided by Giovanni Cardone, Imaging Facility, MPI of Biochemistry), which uses Otsu's method to find the largest connected membrane area. If not stated otherwise, the eGFP channel was used for membrane detection. As a result of the *ImageJ* script, channel-wise pixel values representing photon counts confined to the membrane mask were saved to TXT files for subsequent analysis. For the microscope and imaging settings used, lateral chromatic aberration was negligible as the displacement to match membrane masks between different color channels was mostly below one pixel in both x- and y-direction. To yield fluorescence intensities, photon counts in each channel were averaged and divided by the pixel dwell time. The average fluorescence intensities

are then divided by the molecular brightness of the corresponding fluorophore determined by FCS in solution to retrieve the number of emitting molecules. For eGFP and IL-4-ATTO647N particle numbers, the number of molecules were rescaled by a factor of $1/(1 - p_{nf})$ to account for a fraction of non-fluorescent labels. We used $p_{nf} = 0.20$ for eGFP[54] and $p_{nf} = 0.32$ for the partially labeled IL-4-ATTO647N (see section *Recombinant IL-4-ATTO647N*). The click efficiency and IL-4 occupancy were then defined as the ratio of Alexa568-tet1 to eGFP particles and IL-4-ATTO647N particles to eGFP particles, respectively.

To estimate receptor densities in the plasma membrane, eGFP particle numbers were rescaled to an elliptical cross-section ($A_{eff} = \pi w_0 z_0$) as determined by the confocal observation volume calibration.

In Fig. 5b, e, f, measured click efficiencies of the IL-4-occupied state ($CE'_{occ}$) were rescaled linearly to take into account the varying degree of IL-4 bound to the receptor mutants ($F_{occ}$) at 10 nM IL-4-ATTO647N (Fig. 2e).

$$CE_{occ} = \left(CE'_{occ} - \left(1 - F_{occ}\right)CE\right)/F_{occ}. \tag{6}$$

## BCRI−calibration of the confocal observation volume

To assess microscope performance and the confocal observation volume, FCS measurements of 25 nM ATTO488-COOH (ATTO-TEC), Alexa568-COOH (Thermo Fisher Scientific) and ATTO655-COOH (ATTO-TEC) in ultrapure water were performed before and after imaging. The confocal detection volume was positioned 15 μm above the glass surface (LabTek #1.5; Thermo Fisher Scientific) and the correction collar of the objective was manually adjusted by maximizing the photon counts to account for varying glass thicknesses. The beam waist radius $w_0$ of the confocal detection volume for each of the three laser lines (488 nm, 561 nm, and 633 nm) was determined by the diffusion time $\tau_{D,st}$ and the translational diffusion coefficient $D_t$ of the freely diffusing standard dyes via the relation $w_0 = \sqrt{4D_t\tau_{D,st}}$. The detection volume is then defined as $V_{eff} = \pi^{3/2}w_0^2 z_0$, where the elongation of the Gaussian detection volume along the optical axis, $z_0$, was retrieved by fitting the axis ratio $AR = z_0/w_0$. Once $w_0$ was determined, unknown diffusion coefficients of protein samples, e.g. receptors in the plasma membrane, were calculated by $D_t = w_0^2/4\tau_D$, where $\tau_D$ denotes the measured diffusion time of the molecule of interest.

To account for elevated temperatures due to laser-dependent heating of the sample, diffusion coefficients of 400 μm² s⁻¹ (ATTO488-COOH), 370 μm² s⁻¹ (Alexa568-COOH) and 426 μm² s⁻¹ (ATTO655-COOH) at 25 °C[55] were adjusted to 28 °C[56]. The diffusion coefficient of Alexa568-COOH was determined by comparing its diffusion time to the diffusion of rhodamine B, for which a diffusion coefficient of 450 μm² s⁻¹ at 25 °C was reported[55].

## BCRI−determination of excitation power density

The total laser power $P_{tot}$ for each laser line (488 nm, 561 nm, and 633 nm) and a range of laser attenuation values was measured before and after imaging with a microscope-slide power meter (S170C, Thorlabs) behind the objective lens. The average $P_{tot}$ of individual imaging sessions was then fitted linearly as a function of time to monitor laser ageing and decreasing power output over the course of the project. The excitation power density $P_0$ in the focal spot was defined as $2P_{tot}/(\pi w_0^2)$, with a beam waist radius $w_0$ being obtained from the confocal observation volume calibration.

## BCRI−determination of molecular brightness with FCS

Fluorophores used for imaging (eGFP, Alexa568-tet1, trisNTA-Alexa647, IL-4-ATTO647N) were diluted to 100 nM in either PBS (pH 7.4) in case of eGFP or in air buffer. For dyes coupled to a tetrazine, a 500-fold excess of BCNK was added to the solution and equilibrated for 30 min.

FCS measurements were performed for a range of laser powers with beam path, filter and pinhole settings matching the ones used for imaging, and acquisition times ranging from 3 to 15 min depending on laser power. Autocorrelation functions were fitted with a 3D diffusion model,

$$G(\tau) = \frac{1}{N}G_{3D}(\tau). \tag{7}$$

$$G_{3D}(t) = \left(1 + \frac{\tau}{\tau_{3D}}\right)^{-1}\left(1 + \frac{\tau}{AR^2\tau_{3D}}\right)^{-\frac{1}{2}}. \tag{8}$$

where $N$ denotes the number of particles in the detection volume, $\tau_{3D}$ the diffusion time of freely diffusing particles and $AR$ the axis ratio defined as $z_0/w_0$. The molecular brightness is given by the ratio of the fluorescence intensity $F$ to the particle number $N$ ($CPP = F/N$).

## Calibration of concentrations in titration assays of trisNTA-Alexa647 and IL-4-ATTO647N

After 30 min equilibration time, unbound ligand in bulk solution (trisNTA-Alexa647 or IL-4-ATTO647N) was assessed by FCS measurements at three different positions in-between cells. Recordings were split into 6 runs à 30 s, and runs showing slow intensity drifts or intensity spikes were discarded. Autocorrelation functions were fitted with a 3D diffusion model (Eqs. 7 and 8). Erroneous fits, occurring especially at low concentrations due to noisy data, were identified by inconsistent *CPP* values as compared to the molecular brightness references of trisNTA-Alexa647 or IL-4-ATTO647N (see section *BCRI− determination of molecular brightness with FCS*) and filtered out. Particle numbers $N$ were averaged position-wise and converted to local concentrations $c$ via $c = N/(N_A V_{eff})$, where $N_A$ is the Avogadro constant and $V_{eff}$ is the effective observation volume determined from the diffusion of ATTO655-COOH (see section *BCRI−calibration of the confocal observation volume*). In case of IL-4-ATTO647N, particle numbers were rescaled by a factor of $1/(1 - 0.32)$ to account for 32% non-labeled IL-4.

By means of the actual ligand concentrations in the supernatant, binding curves were modeled with the Hill equation

$$\theta = \frac{A}{1 + \left(c_0/c\right)^n} + A_0. \tag{9}$$

in a non-linear least-squares fit (*scipy.optimize.least_squares* function, version 1.6.2), where $\theta$ is the fraction of receptors bound by ligand, $A$ the saturating fraction, $A_0$ the background level, $c_0$ the dissociation constant, $c$ the concentration of ligand and $n$ the hill coefficient. To include the uncertainties of both $\sigma_x$ (concentration) and $\sigma_y$ (occupancy) in parameter estimation, fitting was repeated 10,000 times in a Monte Carlo approach. Each time, random numbers from Gaussian distributions of width $\sigma_x$ or $\sigma_y$ and mean 0 were drawn and added on each data point before fitting. The reported dissociation constants represent mean ± SD of the 10,000 resamples.

## Statistics & reproducibility

No statistical method was used to predetermine sample size. No data were excluded from the analyses. Statistical significance of IL-4-induced shifts in click efficiency was tested in Python with two-sided Welch's *t*-test to take unequal population variances and sample sizes into account (*scipy.stats.ttest_ind* function, version 1.6.2).

For correlative analyses of CE and differential CE, Pearson correlation coefficients $r$ and two-tailed *p*-values were calculated in Python (*scipy.stats.pearsonr* function, version 1.6.2). To include the uncertainties $\sigma$ of CE and differential CE, 10,000 Monte Carlo simulations were performed. Each time, a random number from a Gaussian distribution of width $\sigma$ and mean 0 was drawn and added on each data

point before calculating $r$ and $p$. The reported $r$ and $p$-values in Figs. 5b, e, f and Supplementary Fig. 29 represent mean ± SD of the 10,000 resamples.

## Analysis of ligand-dependent structural backbone deformations

Crystal structures of the IL-4Rα ectodomain were taken from the type 2 IL-4R complex, either complexed with IL-4 (PDB: 3BPN) or IL-13 (PDB: 3BPO) as a natural ligand[27]. Since the affinity of IL-13 towards the IL-4Rα chain is negligible[27,30], the latter served as a proxy for unliganded IL-4Rα chain, for which no crystal structure was available. Since the overall orientation of the two fibronectin type III domains (D1 and D2) is uncertain in their native state at the plasma membrane, we focused on local deformations. A sequence alignment followed by a structural superposition was performed without further arguments using the command *align* (PyMOL, version 1.8.4.2). For the linker positions E94B and K97B a global alignment of both domains D1 and D2 (amino acids #2–#196) was performed. For all other receptor mutants, the domains D1 (amino acids #2–#96) and D2 (#98–#196) were aligned separately, thus minimizing orientational mismatch. To quantify structural deformations, the root-mean-square deviation (RMSD) of the Cα-atom anchoring BCNK including ±1, ±2, and ±3 adjacent Cα-atoms ($RMSD_0$, $RMSD_1$, $RMSD_2$, $RMSD_3$) were exported using the command rms_cur.

In Fig. 5e, Fig. 5f and in Supplementary Fig. 29, the limits of the respective secondary y-axis on the right were chosen such that the average spread of differential CE and RMSD is equal.

## Molecular dynamics simulations

All-atom molecular dynamics simulations were carried out with the *AMBER* suite of programs (version 20)[57]. Set up of structures and geometries: BCNK was constructed with *MOLDEN* (version 5.9.2)[58] and minimized with *GAUSSIAN* (version 09)[59] applying b3lyp/3-21g* level of theory. Optimized coordinates were saved in mol2 format and used to generate *AMBER* parameters with the help of programs *antechamber* and *prepgen* first neglecting partial charges, then, after quick *sander* minimization, including the generation of *bcc* charges[60]. Resulting parameters were fine-tuned to be maximally compatible with standard protein residues of the *ff14SB* force field[61] also making use of *gaff* parameters[62]. Having been designed as pseudo amino acid residue, the new BCN type could substitute any particular site in a given PDB file by converting the name of the amino acid in question to BCN and retaining just atoms of the peptide bond, which however needed to be renamed as N-C2-C3-O3 rather than the standard N-CA-C-O (an example is given in Supplementary Software SI_comp_bio/a). Counter ions, occasionally *N*-Acetylglucosamine and explicit water were described by respective force fields, *GLYCAM_06j-1*[63], *ionsjc_tip3p*[64] and *tip3p*[65]. All structures were neutralized with $Na^+$ or $Cl^-$ ions and embedded in boxes of explicit water (TIP3P) using a minimum distance of 14 Å between any protein atom and the border of the box. Systems studied comprise the wt complex IL-4R/IL-4 (PDB: 3BPN)[27] and single amino acid substitutions with BCNK, 3bpn_T18B, 3bpn_K22B, 3bpn_S30B, 3bpn_S44B, 3bpn_K87B, 3bpn_E94B, 3bpn_K97B, 3bpn_E189B thereby including/neglecting IL-13Rα1 and IL-4 parts depending on the query at hand. *VMD* (version 1.9)[66] was used for graphical control.

## Minimization, equilibration, production MD

500 steps of steepest descent minimization were carried out with program *sander* followed by another 500 steps of conjugate gradient minimization using a cutoff of 12 Å. Minimized structures were heated to 300 K within 100 ps of equilibration MD using Langevin dynamics in the NTV ensemble (program *pmemd.MPI*). Particle mesh Ewald summation was applied (cutoff radius of 12 Å, time step 2 fs) and bonds involving H-atoms were considered by the SHAKE algorithm[67]. Continuing with endpoint structures another 100 ps of equilibration MD were sampled at identical conditions except that pressure coupling

was introduced (ntb = 2, ntp = 1, barostat = 2, taup = 2.0). This was followed by another 500 ps of equilibration MD to switch from program *pmemd.MPI* to program *pmemd.cuda*[68] at otherwise identical conditions. Equilibrated systems were taken up for production MD at identical simulation conditions. Overall sampling time in production runs was 250 ns and the anticipated number of saved snapshots in individual MD trajectories was 10,000 (example scripts and input files are provided in Supplementary Software folder SI_comp_bio/b).

## Post-processing of trajectories—BCNK catchboxes

To explore the space of potential BCNK reaction partners a cuboid box (13.5 Å x 4.5 Å x 3.5 Å) was constructed around the central carbon-carbon triple bond of the ring system (atoms C14 and C15 already defining the new local x-axis of the catchbox) for each of the structural snapshots in individual MD trajectories. All non-BCNK atoms in the IL-4R/IL-4 complex were examined as to whether or not they fall into the catchbox. Resulting domains could designate the "outreach potential" of BCNK reactivity in terms of steric hindrance (example script with subset of structural snapshots provided in Supplementary Software folder SI_comp_bio/c). Moreover, catchbox atoms were analyzed with respect to frequency of detection (Supplementary Fig. 24) and corresponding surface patches were visualized on the MD average structure.

## Post-processing of trajectories—average structures

Average structures were computed from MD trajectories using *cpptraj* guidelines provided at https://ambermd.org/tutorials/basic/tutorial3/section6.htm. This was followed by a quick minimization to clean bare average structures from geometric distortions affecting sites of enhanced mobility (an example script is given in Supplementary Software folder SI_comp_bio/d).

## Post-processing of trajectories—rotary motion of BCNK ring system

Two vectors located inside the BCNK ring system were extracted for each structure stored to the MD trajectories. These vectors were defined from the atoms C11→C15 and C18→C14 of the BCN residue and program *cpptraj* was employed (see the example provided in Supplementary Software folder SI_comp_bio/e). Subsequently every vector pair was taken into account for calculation of the cross product and the resulting normal vectors were normalized and saved. Cartesian coordinates of each normal vector were then expressed in spherical polar coordinates and corresponding distributions of azimuthal and polar angles were visualized with the help of kernel density estimates. Two-dimensional maps of polar and azimuthal angles were computed with the Python package *scikit-learn* (version 0.24.2, *sklearn.neighbors.KernelDensity*) using a Gaussian kernel. The optimal bandwidth of the kernel was estimated for each mutant by cross-validated grid-search over a parameter grid of 100 logarithmically spaced bandwidths between 0.1 and 10 (*sklearn.model_selection.GridSearchCV*, Leave-One-Out cross-validator). The resulting 2D kernel densities were plotted (*matplotlib*, version 3.3.4) as filled contours with eight logarithmically spaced levels. Probability densities below $10^{-6.5}$ were displayed in white.

## Post-processing of trajectories—B-factors with respect to the average structure

The MD simulation considering the wild-type, IL-4R/IL-4 (PDB: 3BPN), was analyzed with program *cpptraj* and an average structure was derived. Next, this average structure (not minimized further) formed a reference for B-factor analysis quantifying atomic root-mean-square fluctuations of key residues (destined for BCNK substitution) from the reference along the trajectory (only peptide bond atoms C, CA, N, O were taken into account, see the example provided in Supplementary Software folder SI_comp_bio/f). This was repeated in exactly the same

manner now considering just the isolated receptor, IL-4R, devoid of IL-4 for direct quantitative comparison (see Supplementary Software SI_comp_bio/f/no_il4).

### Calculations of solvation-free energy ($\Delta G^{solv}$)

MD trajectories were split into 10 sub-trajectories from which average structures were computed as described and subjected to Poisson Boltzmann calculations[46] within the framework of the Polarizable Continuum Model (PCM)[48]. Here, the net solvation-free energy (SFE), $\Delta G^{solv}$, is decomposed into a polar term, $\Delta G^{pol}$, accounting for the polarization of the solvent (see sketched field lines and residual polarization charges in Fig. 5a) and two nonpolar terms, $\Delta G^{cav}$ and $\Delta G^{disp}$, which take into account creation of an empty void inside the solvent (white domain in Fig. 5a) of equal size and shape to the solute[69], and attractive van der Waals interaction occurring at the solute/solvent boundary[70,71]. The program *POLCH* (version 2.3)[49] was used and results are summarized in Supplementary Table 7. Erroneous calculations with $\Delta G^{solv}$ values more than three standard deviations away from the mean of each mutant were disregarded.

In Fig. 5b, the offset and limit of the secondary y-axis on the right was chosen such that the average spread of CE and SFE is equal.

### Reporting summary

Further information on research design is available in the Nature Portfolio Reporting Summary linked to this article.

## Data availability

Raw microscopy data generated during the study are available on request due to the large file size, requests should be made to the corresponding author and will be answered within 2 weeks. The following publicly available datasets from the Protein Data Bank were used in the study: 1IAR, 3BPN, 3BPO, 3BPL. Source data are provided with this paper.

## Code availability

The *ImageJ* plugin for cell membrane segmentation of multi-channel confocal images is deposited in the Max Planck data service Edmond (https://doi.org/10.17617/3.YDSJ2C). Custom Python code for subsequent analysis and plotting is available on request from the corresponding author and will be provided within 2 weeks. Materials and code regarding molecular dynamics simulations and post-processing are provided as a Supplementary Software file (SI_comp_bio.zip).

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

## Acknowledgements

We thank S. V. Mayer and K. Lang for synthesis of custom-designed fluorophore-tetrazine conjugates and for plasmids and initial guidance on amber suppression and labeling experiments. We thank G. Cardone

for implementing the image segmentation script and S. Thomas for preparing fluorescent IL-4. We are grateful to M. Schaper and K. Nakel for cloning procedures, and B. Scheffer for support with cell culture. The computational results presented have been achieved using the Vienna Scientific Cluster (VSC). F.S. acknowledges support and funding granted by J. Plitzko and W. Baumeister. P.Schw. acknowledges funding from the Center for Integrated Protein Science Munich and from the Center for NanoScience. We thank G. Vriend for providing biologically relevant B-factors and helpful discussions.

## Author contributions

F.S. and P.S. performed the experiments and analyzed the data. S.H. performed molecular dynamics simulations and analysis. T.D.M. supervised IL-4 production and labeling and analyzed binding data. P.Schw. supervised FCCS analysis. F.S. and T.W. developed BCRI, T.W. performed structural analysis, and wrote the manuscript. All coauthors reviewed and commented on the manuscript.

## Funding

## Competing interests

The authors declare no competing interests.
