## [Peer Review File · Nature Communications]

Reviewers' Comments:

Reviewer #1:

Remarks to the Author:

his review focuses on the atomic detail simulation, structural and energetic analysis. The manuscript provides a detailed study on the use of click chemistry with bioorthogonal labels to probe cell-surface receptors, indicating that the method is sensitive to the details of the local environment of the label. The method appears to have significant potential and it will be interesting to see what is learnt by applying the approach to other proteins in the future.

The following should be addressed upon revision:

Abstract: The wording of the abstract should be modified so that the conclusions are in line with the results and not overstated.

L30 "solvent polarization to govern BCNK reactivity" - the authors have merely found that "computed solvation free energy correlates with BCNK reactivity".

"capable of discerning structural states of proteins" -> the click efficiency shows some correlation with the relative local deformation of the structure at the site of the label upon binding IL4. It is not clear that the differences in structure are functionally relevant; they likely rather reflect the perturbation of the system by the label.

Similarly, the text of the conclusion should be revised.

p12 around l14, Fig 4b and Fig S21, and methods "post-processing of trajectories: B-factors with respect to the average structure": RMS fluctuations, not RMSD, can be computed from B-factors. The text and the legends and y-axis labels should be changed and the computed quantity checked.

Fig 5 legend: RSMD -> RMSD

Molecular dynamics section of Methods in SI: Why use bcc charges instead of resp charges? The Gaussian calculations have already been done.

Fig. S23: "MD snapshot". The time interval between MD snapshots should be given in the legend.

Reviewer #2:

Remarks to the Author:

I think this is a very interesting manuscript that brings together cross-correlation autocorrelation analysis with confocal imaging to provide novel information on the efficiency of bioorthogonal labelling of particular amino acids on the extracellular elements of a single membrane-passing protein. As a consequence, new information is provided on the conformations adopted in response to ligands. I cannot comment on the molecular dynamic analysis and so will confine my comments to the general approach used to investigate ligand-dependent receptor states.

The brightness-calibrated ratiometric imaging approach seemed to work well for determining the occupancy of bound fluorescent IL-4. THE FCS studies have been very carefully undertaken with a lot of appropriate control data provided throughout. However, I found Figure 2c and 2d a bit difficult to understand because we weren't given the actual concentrations of fluorescent label used. This should be provided. Similarly, rather than the x axes being labelled as dilution steps, the actual concentrations of e.g. IL4-ATT0647N (in Figure 2d) should be given. This would make it easier to relate to Figure 2f. I have the same criticism for Supplementary Figure S11 where again we are not provided with the actual concentrations of trisNTA-Alexa647. In this figure, I was expecting the position of the displacement curve for binding to Lyn-eGFP to be very different from that of IL-4Ra since the fluorescent label should not have access to the His-Tag. Any reason for this? I accept that the level of labelling is very different.

My main concern for the brightness calibrated ratiometric approach is that it is based on the molecular brightness of each label monitored in free aqueous solution. What impact does the environment of the fluorescent species have on the molecular brightness observed? For example,

some dyes are quenched in aqueous environments compared to their brightness in lipid environments. This would presumably have an effect on correction for number of fluorescent particles in a membrane. Have FCS studies been performed with different expression levels of IL4R to check that the brightness of the bound IL4-ATT0647N changes as expected based on the BCRI approach? The eGFP label on the intracellular side of the protein gives you a good way to control for this. The same controls could be done for the effect of IL4 expression on the molecular brightness of trisNTA-Alexa647.

Reviewer #3:

Remarks to the Author:

The authors Schultz and coworkers describe in the manuscript the development of methodology combining site specific click labeling with confocal live-cell imaging to provide information about structural states of cell surface receptor. To achieve that, they used 'standard' genetic code expansion to introduce the clickable BCNK nCAA to different positions within the extracellular domain of IL-4Ra. Then they used/developed a Brightness-Calibrated Ratiometric Imaging (BCRI). Finally, detailed investigation on the click efficacy was performed to finally link the different click efficiencies to conformational changes of the protein on the cell surface. The work seems to be well performed and the basic claims are sufficiently supported by experimental details.

Even so I find the claimed generality of the described methodology slightly overstated (example shown for one receptor type) I believe the work could be of interest to the field. In fact, I find the described detailed analysis of the click yield under native conditions directly on the receptor very interesting. This part of the work contains data that could be of interest to the click-chemistry community as well. I am not an expert in confocal microscopy or structural biology and therefore, I do not feel to be in a position to judge the work from that perspective. However, I am familiar with the nCAA and especially with biocompatible labeling applications. I did not find any flaws or inconsistencies in this part of the work and I believe that others will be able to reproduce the results based on the data provided. Instead, I have rather more general comment or suggestion what could be improved. I find the work to be scholarly poorly presented in a sense that it is understandable only for people with rather focused background. As already mentioned, I am not an expert in particular techniques mentioned and used throughout the work, but I find unfortunate that I really had difficulties to go through the manuscript and understand what the authors were trying to achieve. I was especially confused if the methodology was supposed to quantify the click efficiency and analyze different factors that could be instrumental in this regard (position of the nCAA, the microenvironment, solvent exposure/polarity etc.) (which I find very interesting and well performed) or, the main aim was to develop a methodology enabling analysis of subtle structural changes in the protein/receptor as such (which is also an important achievement). To improve this, I suggest including a discussion section in the work, which summarizes the main goals and achievements in a more clear and comprehensible form. I think this could significantly improve the quality of the manuscript and finally, even attract broader readership.

POINT-BY-POINT RESPONSE TO REVIEWER COMMENTS

Manuscript-ID: NCOMMS-22-37432A

We thank the reviewers very much for their interest and the overall supportive responses. Besides the objective to balance our conclusions (reviewer #1 and #3), we invested much efforts to improve the clarity of the text, in particular for the BCRI section (reviewer #2) and some other parts (changes are indicated by red font). The body text has been restructured with a newly written discussion section (reviewer #3). We hope that the finding, goals and achievements are now clearer.

During revision, we noticed incorrect starting conditions in the MD simulation of receptor mutant S44B. The MD simulation has been repeated, leading to almost invisible shifts for the solvation free energy (Fig. 5b, Supplementary Table 6) and an increased rotor flexibility (Supp. Fig. 27). For this reason, we decided to replace S44B with E94B as a better example for confined location of BCNK on the protein surface (Figs. 4d, 4f). In addition, we found accidentally flawed data describing the catchbox analysis which has been corrected (Supp. Fig. 26, Supp. Table 5), and a typo in the formula calculating receptor densities which shifted the x-axis slightly towards higher values (Figs. 3a, Supp. Figs 2 and 14). None of these corrections had an effect on our conclusions. We apologize for this complication.

Reviewer #1 (Remarks to the Author):

This review focuses on the atomic detail simulation, structural and energetic analysis. The manuscript provides a detailed study on the use of click chemistry with bioorthogonal labels to probe cell-surface receptors, indicating that the method is sensitive to the details of the local environment of the label. The method appears to have significant potential and it will be interesting to see what is learnt by applying the approach to other proteins in the future.

Response: We thank the reviewer for the positive assessment of our work. As pointed out correctly, this current work is restricted to the IL-4R α /IL-4 system to establish methodical foundations at sufficient levels of detail. We agree 100% with the reviewer that applying BCRI to additional receptor/ligand systems will be very interesting in the future as this is the sole motivation of the current work.

The following should be addressed upon revision: Abstract: The wording of the abstract should be modified so that the conclusions are in line with the results and not overstated. L30 “solvent polarization to govern BCNK reactivity” - the authors have merely found that “computed solvation free energy correlates with BCNK reactivity”.

Response: Indeed, SFE values mirror variations (!) in absolute click efficiency not the magnitude of click-efficiency *per se*. We propose the following reformulation (main text, lines 31-32):

“Molecular dynamics and continuum electrostatics calculations suggest solvent polarization to determine site-specific variations of BCNK reactivity.”

“capable of discerning structural states of proteins” -> the click efficiency shows some correlation with the relative local deformation of the structure at the site of the label upon binding IL4. It is not clear that the differences in structure are functionally relevant; they likely rather reflect the perturbation of the system by the label. Similarly, the text of the conclusion should be revised.

Response: Again, we thank the reviewer for requesting a more precise language. Since both of the introduced measurable quantities ‘absolute click efficiency’ (correlating with SFE) and ‘differential click efficiency’ (correlating with RMSD) reflect potentially interesting qualities of the target protein, we widened the scope of the concluding sentence in the abstract (main text, lines 35-36): “Thus, click efficiency by itself represents a remarkably informative readout linked to protein structure and dynamics in the native plasma membrane.”

Based on our data, BCNK incorporation causes the major perturbation in bioorthogonal labeling. We see not much influence of the click label on functionality, which is in our case the capability to bind the natural ligand IL-4. To clarify this issue, we included a new paragraph in the discussion dealing with the ‘functional relevance’ of BCNK incorporation (main text, lines 475-487):

“To showcase the abilities of the BCRI approach, we first address functional consequences of the insertion of BCNK at different positions in the extracellular domain of IL-4R α with a single-cell binding assay. As expected, IL-4 binding is mostly abolished when the bulky BCNK locates within the binding epitope, whereas it remains preserved for all non-neutralizing sites. However, quantifications show that BCNK reduces ligand affinity in a site-specific manner. In particular, sites in or close to the activation loop show bimodal distributions of reduced affinity, which hints at strong allosteric effects. Susceptibility towards receptor occupancy in combination with structural vulnerability are a hallmark of conformational changes in signal transduction. It appears quite remarkable that site-specific bioorthogonal labeling in combination with cellular quantifications enable to pinpoint such features within the receptor structure. IL-4 binding assays in the presence of labeled BCNK produced similar results, suggesting that the BCNK incorporation represents the dominant perturbation whereas additional click labeling has minor functional consequences.”

Structural correlations were detected for ligand-dependent ‘differential click efficiency’, a measurable quantity, where site-specific variations linked to solvation of the target protein seem to cancel. We now discuss this remarkable finding in much more detail (main text, lines 544-555):

“Encouraged by the intriguing link between structural topology of the BCNK bearing receptor mutants and absolute CE, we investigated the effect of ligand-induced structural changes. Comparing CE signatures of free and IL-4-bound receptors shows small but distinct shifts, which correlate remarkably well with structural deformations in the extracellular domain of IL-4R α . The sub-angstrom scale of backbone translocations is strikingly small, raising the question of how such a great sensitivity is conceivable. We suppose that BCNK- and IL-4-induced structural changes are largely uncoupled. According to this assumption, structural perturbations induced by BCNK

are independent of receptor occupancy and site-dependent contributions mostly cancel when computing the differential CE. Since IL-4 imposes structural changes of the protein backbone as well, the three-dimensional organization of charge centers is nonetheless affected. Thus, residual changes in CE seem to scale in linear approximation with small BCNK translocations within a quasi-static energy landscape.”

The previous conclusions section has been removed. Instead, we conclude the discussion with a proposal for an experimental scheme that, in our opinion, represents a promising way to ‘discern protein states’. We hope that the final paragraph is now in line with the results and findings. (Main text, lines 557-564):

“The capability to sense conformational changes in the tertiary structure of a target protein in living cells is auspicious. These findings suggest a generic strategy to approach conformational substates at the cell surface (Fig. 6). CE signatures may be regarded as fingerprints that allow to associate protein conformations with certain experimental conditions and hence facilitate to identify molecular constraints that define these substates. Thus, in perspective, bioorthogonal labeling in combination with BCRI could shed new light on dynamic protein transitions involved in signal transduction at the plasma membrane and potentially other dynamic processes in molecular cell biology.”

p12 around 114, Fig 4b and Fig S21, and methods “post-processing of trajectories: B-factors with respect to the average structure”: RMS fluctuations, not RMSD, can be computed from B-factors. The text and the legends and y-axis labels should be changed and the computed quantity checked.

Response: We apologize for having provided inaccurate information and corrected the affected passages:

- a) The respective sentence referring to Fig. 4b has been adjusted (main text, lines 330-332):
“Computed B-factors from atomic fluctuations of the peptide bond at potential BCNK incorporation sites showed larger magnitudes as those extracted from experimental B-factors [...]”
- b) Atomic displacement parameters in Fig. 4b and Supp. Fig. 21 were changed from RMSF to B-factors. In Supp. Fig. 21, a typing error in the calculation of B-factors from anisotropic atomic displacement parameters was corrected as well.
- c) Legends were edited in Fig. 4b (main text, lines 348-349):
“B-factors of atoms forming the BNCK peptide bonds from MD simulation of IL-4R α (PDB: 3BPN) show reduced motion at key positions of IL-4 ligand binding.”
and in Supp. Fig. 21 (SI, lines 720-722):
“Experimental B-factors of the C α atom of corresponding BCNK junctions in the protein backbone (mature numbering; Uniprot P24394-1) extracted from crystal structures (PDB: 1IAR, 3BPL, 3BPO, 3BPN)^{14, 35}”
- d) Correction in the respective paragraph “Post processing of trajectories—B-factors with respect to the average structure” (SI, lines 380-382):

“Next, this average structure (not minimized further) formed a reference for B-factor analysis quantifying atomic root-mean-square fluctuations of key residues (destined for BCNK substitution) [...]”

Fig 5 legend: RSMD -> RMSD

We thank the reviewer for spotting this typo. The misspelling has been corrected (main text, lines 406-407)

“[...] represents the translocation of the BCNK junction without neighboring C α atoms. (d) RMSD of the BCNK junction at different incorporation sites.”

Molecular dynamics section of Methods in SI: Why use bcc charges instead of resp charges? The Gaussian calculations have already been done.

This is a very relevant remark. From our perspective, there is no particular reason to prefer bcc charges over direct RESP charges; the use of bcc charges was rather a choice out of convenience related to recently established protocols. In our opinion, it appears much more important to provide maximum levels of compatibility to amino-acid-class residues when assigning partial charges to the new BCN residue. For the subject matter expert:

- a) starting from GLU as template we set up an initial BCN skeleton from within xleap
- b) continue structure-editing within MOLGEN and introduce protection groups, -CO-CH₃ and -NH-CH₃ for open valencies of atoms forming the peptide bond
- c) b3lyp/3-21* optimize the geometry
- d) iteratively refine the geometry based on TINKER minimizations using MM3 parameters
- e) have the intermediate BCN structure cross-checked and confirmed from literate chemistry fellows
- f) do a second b3lyp/3-21* optimization with the confirmed intermediate BCN structure
- g) take up the minimized geometry from the previous step in MOLGEN and save as *.mol2 file and change residue name to BCN
- h) antechamber/prepgen translate the *.mol2 file to AMBER-internal formats, bcnk.prepgen.pdb and bcnk.prepi without assigning partial charges (again replacing MOL residue names with BCN)
- i) load the *.prepi *.pdb files from the previous step into xleap and define missing GAFF-parameters based on repeated saveamberparm-attempts until all dependencies are resolved
- j) sander-minimize the system from the previous step (lacking any partial charges at this point) and take up the optimized geometry and save another *.pdb file, bcnk.xleap.2.pdb and visually check it within VMD
- k) repeat the few set-up steps for defining a new residue (BCN), however, starting from the just optimized structure and this time including the calculation of bcc partial charges considering overall neutrality
- l) load the *.prep *.frcmod *.pdb files resulting from the previous step into xleap and do fine-tuning of force field parameters (name, retype, head/tail atoms, retype, name,

connect0/1 atoms) and manually adjust partial charges of peptide-bond-forming atoms in the new BCN residue to be identical to those of residue ALA, in particular,

i.	NAME	TYPE	CHARGE	ELEMENT
ii.	N	n	-0.4157	N -0.0194 ==> -0.4351
iii.	H27	h	0.2719	H
iv.	C2	cx	0.0337	C
v.	C3	c	0.5973	C
vi.	O3	o	-0.5679	O -0.0142 ==> -0.5821
vii.	H	h1	0.0823	H

m) calculate the net charge in this modified BCN residue (0.0336) and thus slightly increase magnitudes of partial charges on N and O atoms by -0.0194 and -0.0142 to counterbalance the non-zero net charge and save the final *.prep file for subsequent usage

To streamline the description of individual steps in the above procedure, we provided the resulting force field parameters in the Supplementary ZIP file (SI_comp_bio/a/bcnk.frmod and SI_comp_bio/a/bcnk.4.prep) and mentioned major methodological steps in the SI.

Fig. S23: “MD snapshot”. The time interval between MD snapshots should be given in the legend.

The requested time interval has been provided in the legend of Supp. Fig 23 (SI, line 746):
“The time interval between MD snapshots is 25 ps”.

Reviewer #2 (Remarks to the Author):

I think this is a very interesting manuscript that brings together cross-correlation autocorrelation analysis with confocal imaging to provide novel information on the efficiency of bioorthogonal labelling of particular amino acids on the extracellular elements of a single membrane-passing protein. As a consequence, new information is provided on the conformations adopted in response to ligands. I cannot comment on the molecular dynamic analysis and so will confine my comments to the general approach used to investigate ligand-dependent receptor states.

Response: We thank the reviewer for the positive feedback on our work.

The brightness-calibrated ratiometric imaging approach seemed to work well for determining the occupancy of bound fluorescent IL-4. THE FCS studies have been very carefully undertaken with a lot of appropriate control data provided throughout. However, I found Figure 2c and 2d a bit difficult to understand because we weren't given the actual concentrations of fluorescent label used. This should be provided. Similarly, rather than the x axes being labelled as dilution steps, the actual concentrations of e.g. IL4-ATT0647N (in Figure 2d) should be given. This would make it easier to relate to Figure 2f. I have the same criticism for Supplimentary Figure S11 where again we are not provided with the actual concentrations of trisNTA-Alexa647.

Response: We certainly share the reviewer’s opinion that actual concentrations of ligands in Fig. 2c and d are much more informative than ‘dilution steps’. We initially refrained from this idea because it poses additional complications that extend the canonical BCRI protocol. Titrations in the nanomolar concentration range are not reliable and concentrations cannot be calculated based on stock solutions and dilution factors. Adsorption of the ligand-dye-conjugates to pipettes, walls of vials, and even the glass support of the measurement chambers causes unpredictable losses. Thus, the active compound participating in the local equilibrium at the cell surface must be determined by an extra series of measurements in the supernatant.

To reach at true binding curves, we performed FCS measurements in the supernatant for each dilution step, calibrated the x-axis and fitted the BCRI data with a 1:1 model for the fractions ligand bound. We were pleased to see that the determined dissociation constants are in excellent agreement with literature for both samples (IL-4 s well as trisNTA). The following changes have been implemented:

The methods section now contains a description of how to combine BCRI and FCS in the supernatant (SI, lines 252-277):

“Calibration of concentrations in titration assays of trisNTA-Alexa647 and IL-4-ATTO647N

After 30 min equilibration time, unbound ligand in bulk solution (trisNTA-Alexa647 or IL-4-ATTO647N) was assessed by FCS measurements at three different positions in-between cells. Recordings were split into 6 runs à 30 s, and runs showing slow intensity drifts or intensity spikes were discarded. Autocorrelation functions were fitted with a 3D diffusion model (equations 7 and 8). Erroneous fits, occurring especially at low concentrations due to noisy data, were identified by inconsistent CPP values as compared to the molecular brightness references of trisNTA-Alexa647 or IL-4-ATTO647N (see section *BCRI—determination of molecular brightness with FCS*) and filtered out. Particle numbers N were averaged position-wise and converted to local concentrations c via $c = N/(N_A V_{\text{eff}})$, where N_A is the Avogadro constant and V_{eff} is the effective observation volume determined from the diffusion of ATTO655-COOH (see section *BCRI—calibration of the confocal observation volume*). In case of IL-4-ATTO647N, particle numbers were rescaled by a factor of $1/(1-0.32)$ to account for 32% non-labeled IL-4.

By means of the actual ligand concentrations in the supernatant, binding curves were modeled with the Hill equation

$$\theta = \frac{A}{1 + (c_0/c)^n} + A_0 \quad (8)$$

in a non-linear least-squares fit (*scipy.optimize.least_squares* function, version 1.6.2), where θ is the fraction of receptors bound by ligand, A the saturating fraction, A_0 the background level, c_0 the dissociation constant, c the concentration of ligand and n the hill coefficient. To include the uncertainties of both σ_x (concentration) and σ_y (occupancy) in parameter estimation, fitting was repeated 10,000 times in a Monte Carlo approach. Each time, random numbers from Gaussian distributions of width σ_x or σ_y and mean 0 were drawn and added on each data point before fitting. The reported dissociation constants represent mean \pm SD of the 10,000 resamples.”

Figure panels 2c and 2d as well as Supp. Fig. 12 were updated accordingly with a new results section (main text, lines 196-203):

“To push cellular quantification even further, we determined the concentration of ligands in the supernatant by single-color FCS. Since only active binding partners at the plasma membrane participate in the equilibrium, it is necessary to account for compound losses during titration. For a given concentration of free ligand, the fraction of bound receptors as determined by BCRI can be approximated with the Hill equation (Supplementary Equation 8). The dissociation constants for trisNTA (34 ± 8 nM) and IL-4 (0.14 ± 0.04 nM) were in excellent agreement with literature^{28,39}, highlighting the power of BCRI for bridging cellular image quantification with solution biochemistry.”

In this figure [Supp. Fig. 11], I was expecting the position of the displacement curve for binding to Lyn-eGFP to be very different from that of IL-4Ra since the fluorescent label should not have access to the His-Tag. Any reason for this? I accept that the level of labelling is very different.

Response: This is a very interesting point. We looked into this in more detail and summarize the results in an updated Supp. Fig. 12 a and c (based on former Supp. Fig. 11). Here, we display total cell-bound ligand density for IL-4R α * expressing cells and Lyn-eGFP expressing control cells in comparison. In both cases, non-specific binding to the cell surface is significant and cannot be ignored when modelling the data. However, as the reviewer spotted correctly, the curves for non-specific binding at the plasma membrane (control cells), in particular for trisNTA, show similar shape as compared to receptor binding and can accordingly be fitted by the same Hill equation. We treat non-specific interactions descriptively, as modelling adsorption at the cell surface is clearly beyond the scope of this manuscript. We like to mention that both ligands are positively charged: IL-4 carries many surface exposed lysines leading to an isoelectric point at pH 9, while trisNTA is complexed by divalent Ni²⁺. Thus, both probes bear inherently attractive potential towards the negatively charged cell surface. We speculate that heterogeneous charge distributions at the cell surface lead to non-specific binding sites with broadly distributed affinities. Therefore, residual non-specific binding can even be sensed in the nanomolar concentration range. We mention this scenario in the legend for Supp. Fig. 12 (SI, lines 587-590):

“(a, c) Cell-bound ligand density plotted against the concentration of free ligand in the supernatant as determined by FCS. At higher ligand concentrations, significant ligand binding occurs even for control cells indicating hidden, probably charge-mediated, binding sites in the native plasma membrane. “

To demonstrate that receptor binding is nevertheless specific, we included correlation plots showing the dependency of ligand-associated signals on receptor surface density in the plasma membrane (Supp. Fig. 12b and 12d). As expected, for trisNTA as well as IL-4, ligand binding scales proportionally with the surface expression of the receptor, whereas the correlation vanishes

in either case for control cells expressing inaccessible Lyn-eGFP. As the concentration of hidden binding sites is not affected by varying expression levels of IL-4R α^* , the non-specific background assessed with control cells can be globally subtracted for each titration step. We write in the legend of Supp. Fig. 12b and 12d (SI, lines 590-595):

“(b, d) Correlation analysis for high concentrations of ligand (grey shaded range in a, c). Ligand binding to the plasma membrane and receptor expression levels are highly correlated when expressing IL-4R α^* (filled circles) but uncorrelated for control cells expressing Lyn-eGFP (unfilled squares). Dashed lines represent fits to the Hill equation (a, c) or the relation expected for a 1:1 binding reaction (b, d). Markers and error bars represent mean \pm SD of 20 to 40 cells.”

My main concern for the brightness calibrated ratiometric approach is that it is based on the molecular brightness of each label monitored in free aqueous solution. What impact does the environment of the fluorescent species have on the molecular brightness observed? For example, some dyes are quenched in aqueous environments compared to their brightness in lipid environments. This would presumably have an effect on correction for number of fluorescent particles in a membrane.

This is entirely true and we thank the reviewer for pointing this out. We agree that the molecular brightness cannot be generalized and must be determined for each specific molecular condition of the fluorophore. We now stress this point explicitly in the manuscript (main text, lines 154-158):

“To generate reliable concentration data by image calibration, it is important to determine the exact photon yield reflected by the counts per particle (CPP) for a given fluorescence detection setting. Therefore, the CPP was assessed by FCS measurements both in cells and in solution (Supplementary Fig. 7). Solution measurements can then be used to cover the typical range of excitation power densities applied for imaging (Supplementary Fig. 8).”

For the click label Alexa568-tet1 we have addressed this issue by a comparative FCS experiment showing that Alexa568-tet1 conjugated to BCNK in air buffer, the buffer in which we image cells, shows the same molecular brightness as the click-labeled receptor mutant (Supp. Fig. 7). Since Alexa568 is twofold negatively charged, the flexible BCNK-tetrazine linker (Supp. Fig. 25) seems to provide sufficient distance from the likewise negatively charged plasma membrane. Supp. Fig. 7 has been shifted up in the manuscript and the legend has been modified for improved clarity (SI, lines 507-517):

“**Supplementary Fig. 7. Molecular brightness measured by single-color fluorescence correlation spectroscopy (FCS).** The molecular brightness of Alexa568-tet1 reflected by the average counts per particle (CPP) in the confocal observation volume was measured by single-color FCS either in free solution (orange circles) or at the bottom membrane of HEK293T cells after click labeling of the receptor mutant K97B (grey circles). To match conditions at the plasma membrane, Alexa568-tet1 in solution was pre-incubated with excess amounts of reactive BCNK for conjugation. Linear regression indicates that the molecular brightness of the fluorophore Alexa568 is the same under both conditions. Laser irradiance P_0 was determined from the total laser power P_{tot} and the beam waist radius w_0 of the 561 nm laser line. Markers and error bars represent the mean \pm SD of three positions in the measurement chamber (in solution) or of 6 to 24 cells.”

However, direct validation of the molecular brightness via intracellular FCS is laborious. Especially for reversible binders, increasing amounts of fast diffusing ligand in the supernatant exacerbate extracting the fraction ligand bound when fitting two-component autocorrelation functions. For this reason, we performed the titration series (Fig. 2 c, d), showing that BCRI can reproduce the 100% saturation as expected for the well-known 1:1 stoichiometry of trisNTA and IL-4. Thus, even without modelling a binding curve comprehensively (as it is now provided in Figs. 2c and 2d), measuring the ligand stoichiometry of occupied receptors under saturating conditions represents a valid control for using the correct CPPs in the BCRI approach. We now mention this briefly (main text, lines 193-195):

“We like to note that reproducing the expected saturation levels for a given stoichiometry provide means to validate the correct CPPs used for image calibration via BCRI.”

Not only the environment but also the correct chemical derivative matters when determining the CPP. For example, the molecular brightness of the Alexa568-tet1 is increased upon reacting with BCNK (Lang et al., 2012, *J. Am. Chem. Soc.*, reference 9 in the manuscript). We confirmed this brightness increase with FCS in solution (by about 20%, not shown) and therefore produced the laser excitation-dependent ‘brightness reference plots’ with Alexa568-tet1 conjugated to BCNK. The figure legends of corresponding Supp. Figs. 8 and 11 have been improved to clarify this point (SI, lines 523-530 and lines 567-574, respectively):

“Supplementary Fig. 8. Molecular brightness reference of the labels eGFP and Alexa488-tet1. Average counts per particle (*CPP*) in the confocal observation volume was measured by single-color FCS in dilute solutions of recombinant eGFP and Alexa568-tet1 coupled to BCNK for a range of excitation power densities. For both fluorophores, the *CPP* scales linearly with the laser irradiance (linear fit, black). The increased slope of Alexa568-tet1 is related to superior performance of the Alexa568 fluorophore in terms of photon yield. Laser irradiance P_0 was determined from the total laser power P_{tot} and the beam waist radius w_0 of the respective laser line.”

“Supplementary Fig. 11. Molecular brightness reference of the labels eGFP and trisNTA-Alexa647. Average counts per particle (*CPP*) in the confocal observation volume was measured by single-color FCS in dilute solutions of recombinant eGFP (PBS, pH7.4) and trisNTA-Alexa647 (air buffer) for a range of excitation power densities. For both fluorophores, the *CPP* scales linearly with laser irradiance (linear fit, black). The increased slope of trisNTA-Alexa647 is related to superior performance of the Alexa647 fluorophore in terms of photon yield. Laser irradiance P_0 was determined from the total laser power P_{tot} and the beam waist radius w_0 of the respective laser line.”

We recently revisited the pH sensitivity of eGFP by FCS and absorption spectroscopy conforming the well-known photophysical states (Steiert et al., 2018, *Biophys. J.*; doi: <https://doi.org/10.1016/j.bpj.2018.04.013>). Due to the compact β -barrel structure of eGFP the chromophore is, at neutral pH, mostly shielded from environmental cues which keeps the photon yield stable for a broad range of conditions.

In our hands, the molecular brightness of receptor-fused eGFP in HEK293T cells matches the molecular brightness measured for recombinant eGFP in PBS at pH 7.4. Erroneous CPP values would have caused deviations from the 100% saturation levels in Figs. 2c and 2d as well, indicating that the CPP of eGFP is stable.

Have FCS studies been performed with different expression levels of IL4R to check that the brightness of the bound IL4-ATT0647N changes as expected based on the BCRI approach? The eGFP label on the intracellular side of the protein gives you a good way to control for this. The same controls could be done for the effect of ILR4 expression on the molecular brightness of trisNTA-Alexa647.

This is an interesting idea. Based on the existing data, we do not expect that the brightness of bound IL-4-ATTO647N or trisNTA-Alexa647 depends on the receptor expression levels. This can already be seen in Supp. Fig. 14 where the cellular receptor occupancy is plotted against the IL-4R α * density based on eGFP at the plasma membrane. The distributions remain horizontal for about two orders of magnitude in receptor density, showing that the CPP used globally to calibrate this data set is accurate for all these conditions. Another example provides the newly added correlation plots in Supp. Fig. 12b and 12d showing a slope of about unity for both reversible ligands for a range of receptor densities.

As mentioned above, intracellular FCS under equilibrium conditions is quite cumbersome due to the increasing amounts of unbound ligand in the supernatant during titration. Highly expressing cells are generally problematic for FCS because the fluctuation amplitudes become small. Therefore, we preferred BCRI above intracellular FCS to address this important issue.

Reviewer #3 (Remarks to the Author):

The authors Schultz and coworkers describe in the manuscript the development of methodology combining site specific click labeling with confocal live-cell imaging to provide information about structural states of cell surface receptor. To achieve that, they used 'standard' genetic code expansion to introduce the clickable BCNK ncAA to different positions within the extracellular domain of IL-4Ra. Then they used/developed a Brightness-Calibrated Ratiometric Imaging (BCRI). Finally, detailed investigation on the click efficacy was performed to finally link the different click efficiencies to conformational changes of the protein on the cell surface. The work seems to be well performed and the basic claims are sufficiently supported by experimental details.

Response: We thank the reviewer for this appraisal.

Even so I find the claimed generality of the described methodology slightly overstated (example shown for one receptor type) I believe the work could be of interest to the field.

Response: We take the criticism of some parts being 'slightly overstated' very serious. In part, overstatement may have been result of mere excitement, in particular for the fact that such subtle

changes in tertiary structure could be sensed with conventional microscopy techniques. As a consequence, the manuscript suffered a bias which we have addressed with our new discussion (see also our response to the last point). In addition, we decided to treat Figure 6 as a proposal by concluding at the end of the discussion (main text, lines 557-559):

“The capability to sense conformational changes in the tertiary structure of a target protein in living cells is auspicious. These findings suggest a generic strategy to approach conformational substates at the cell surface (Fig. 6).”

In fact, I find the described detailed analysis of the click yield under native conditions directly on the receptor very interesting. This part of the work contains data that could be of interest to the click-chemistry community as well. I am not an expert in confocal microscopy or structural biology and therefore, I do not feel to be in a position to judge the work from that perspective. However, I am familiar with the ncAA and especially with biocompatible labeling applications. I did not find any flaws or inconsistencies in this part of the work and I believe that others will be able to reproduce the results based on the data provided.

Response: We thank the reviewer for recognizing the value of this part. We definitely share the vision that BCRI may represent a versatile tool for the click chemistry community even without the necessity of diving into protein structure and dynamics. To stress this point, the new discussion provides a short synopsis about click labeling at the cell surface with a focus on contrasting behavior as compared to labeling in free solution (main text, lines 489-512):

“Since methods for cellular quantifications were limited, effects of varying click labeling conditions at the cell surface have not yet been systematically addressed. Indeed, for iEDDAC employing the BCNK-tetrazine pair, we observe a much lower degree of labeling for cell-bound receptors as compared to free solution⁹. Tetrazine association follows a comparatively slow dual phase kinetics for which saturation is never reached. In addition, the reaction rates show a sublinear concentration dependence. Thus, click labeling at the cell surface cannot be modelled by diffusion-limited pseudo-first order kinetics. It seems the complicated cell topology comprised by a ruffled plasma membrane, a charged glycocalyx layer and the heterogeneous spatial distribution of cell-bound receptors, require a more sophisticated description of the diffusion part. However, the click reaction still remains pseudo-first order in the sense that local receptor-bound BCNK concentrations are low enough to have negligible effect on the reaction rate. This thermodynamic feature allows determination of a representative cell-average by measuring a relatively small number of hand-selected cells that show suitable signal levels for confocal imaging.

Environmental parameters for iEDDAC-mediated conjugation like tetrazine concentration, incubation time, and temperature were mainly limited by abnormal cellular phenotypes and our optimized standard labeling conditions are in agreement with reported protocols^{9, 13, 18, 19}. Importantly, the fraction of click-labeled receptors measured for one and the same receptor mutant in a series of experimental repetitions using standardized conditions revealed a high degree of reproducibility. While the surface density of the GCE-expressed, BCNK-bearing receptors varied over three orders of magnitude, the CE measured under identical conditions remained stable within a few percent of error. Thus, above the background of instrumental (BCRI) and procedural (cell preparations) noise, site-specific variations of CE become meaningful.”

Instead, I have rather more general comment or suggestion what could be improved. I find the work to be scholarly poorly presented in a sense that it is understandable only for people with rather focused background. As already mentioned, I am not an expert in particular techniques mentioned and used throughout the work, but I find unfortunate that I really had difficulties to go through the manuscript and understand what the authors were trying to achieve. I was especially confused if the methodology was supposed to quantify the click efficiency and analyze different factors that could be instrumental in this regard (position of the ncAA, the microenvironment, solvent exposure/polarity etc.) (which I find very interesting and well performed) or, the main aim was to develop a methodology enabling analysis of subtle structural changes in the protein/receptor as such (which is also an important achievement). To improve this, I suggest including a discussion section in the work, which summarizes the main goals and achievements in a more clear and comprehensible form. I think this could significantly improve the quality of the manuscript and finally, even attract broader readership.

Response: We are very grateful for this suggestion. While writing the discussion section, we recognized that the results part could be restructured accordingly. The subheadings of the manuscript have been changed to outline the distinct steps made during this project:

- 1) Introduction of the experimental setup (main text, line 80):
“Multi-Color Labeling of Receptors”
- 2) Introduction of the quantification method (main text, line 147):
“Cellular Quantification by Brightness-Calibrated Ratiometric Imaging (BCRI)”
- 3) Validation of receptor targets (main text, line 205):
“Functional Consequences of BCNK Incorporation”
- 4) Characterization of the click reaction (main text, line 239):
“Conditions for Click Labeling at the Cell Surface”
- 5) Investigation of different incorporation sites (main text, line 314):
“Site-Dependent Variations in Click Efficiency”
- 6) Investigation of ligand-dependent structural states (main text, line 412):
“Ligand-Dependent Variations in Click Efficiency”

Our central aim was simply to comprehensively understand the variations in click efficiency as displayed in Fig. 4a. While results section 1 has merely introductory character, sections 2-6 are consecutively linked, which means the results of each section provides a necessary basis for tackling the next problem. As such, we found cutting away some of the parts left the story incomplete and we ended up trying to meet all of the goals the reviewer outlined so nicely in his report. We hope in this form it is now comprehensible for a broader readership, which may encompass very different angles and interests within the scope of sections 2-6.

Reviewers' Comments:

Reviewer #1:

Remarks to the Author:

This review focuses on the atomic detail simulation, structural and energetic analysis. The authors have addressed the points raised. The manuscript is improved but some of the analysis still has issues. I noticed the following which should be addressed upon revision:

Fig 5: RMSD is always positive by definition (as given in Table S7). In (e) and (f) it is shown as having the same sign as the click efficiency (negative or positive) but this is not correct. A different representation should be used, e.g. xy plot. Its not obvious how this issue is dealt with in computing the r and p values.

Fig. S21: Last sentence of legend: For which of the structures were no isotropic B-factors available and the mentioned conversion of anisotropic B-factors was necessary? The resolution of the structures is not sufficient for determination of anisotropic B-factors in general and the mentioned pdb files contain isotropic b-factors.

Fig S22:

color-ramp from green to magenta: only green spheres can be seen.

SI:

The description of how the charges were derived given in the reponse should be put in the README in the SI.

It would be more useful if (instead of large movie files) the authors provided pymol/vmd sessions for the structures and deposited the trajectories to make them readily available to the reader.

Reviewer #2:

Remarks to the Author:

As mentioned in my original report, this is a very interesting manuscript that brings together cross-correlation autocorrelation analysis with confocal imaging to provide novel information on the efficiency of bioorthogonal labelling of particular amino acids on the extracellular elements of a single membrane-passing protein. As a consequence, new information is provided on the conformations adopted in response to ligands. I cannot comment on the molecular dynamic analysis and have confined my comments to the general approach used to investigate ligand-dependent receptor states.

The authors have adequately dealt with the major points that I raised previously. The provision of actual concentrations in Figure 2c and d have improved the understanding of the manuscript greatly. I have no other concerns. The changes to the Supplementary figures are also welcome.

Reviewer #3:

Remarks to the Author:

The revised manuscript by F. Steiert and co-workers addresses my original comments. My major concern was the poor scholarly presentation of the otherwise very interesting results. This has been resolved by including a discussion part that briefly summarizes the major findings of the work and is written in a more general language that is now much more enjoyable to read. I find that the additional changes improved the overall quality of the work and I do not have any further comments, suggestions or requests. Congrats to all the authors for a nice work.

POINT-BY-POINT RESPONSE TO REVIEWER COMMENTS

Manuscript-ID: NCOMMS-22-37432B

Reviewer #1 (Remarks to the Author):

This review focuses on the atomic detail simulation, structural and energetic analysis. The authors have addressed the points raised. The manuscript is improved but some of the analysis still has issues. I noticed the following which should be addressed upon revision:

Response: We are pleased that all of the previous points have been addressed and thank the reviewer for careful re-evaluation of the manuscript.

Fig 5: RMSD is always positive by definition (as given in Table S7). In (e) and (f) it is shown as having the same sign as the click efficiency (negative or positive) but this is not correct. A different representation should be used, e.g. xy plot. Its not obvious how this issue is dealt with in computing the r and p values.

Response: This comment concerns a central point. In the previous version we found it intuitive to adopt the experimentalists' view, experiencing a gain or loss in click efficiency upon changing conditions. However, since it is user-defined which condition is taken for reference, signs for differential CE are arbitrary. The situation is similar to structural shifts, which can lead, depending on parametrization, into positive and negative directions. The latter complication is eliminated by using RMSD values.

In light of the reviewer's comment, we presume it is mathematically more rigorous to display absolute values of differential CE, thereby supporting visually the correlation of two genuine positive observables as they have anyway been used to compute the *r*- and *p*-values (values therefore remain the same). We have replaced consistently panels in Fig. 5e, 5f and Supplementary Fig. 29.

Fig. S21: Last sentence of legend: For which of the structures were no isotropic B-factors available and the mentioned conversion of anisotropic B-factors was necessary? The resolution of the structures is not sufficient for determination of anisotropic B-factors in general and the mentioned pdb files contain isotropic b-factors.

Response: We apologize for our brevity on this complicated issue. The B-factors deposited in the PDB files of the four structures were derived with different refinement models and are therefore not directly comparable. In particular, B-factors in PDB-files of the Garcia lab (3BP*) are 'residual' since TLS contributions of groups of atoms were not included in the modelling with REFMAC at that time. Independent of the resolution at which the structure has been solved, this can be corrected for. This issue was comprehensively addressed by W.G. Touw and G. Vriend (Supplementary Reference 36, <https://doi.org/10.1093/protein/gzu044>), where they introduce a public database

providing PDF-files in which B-factors were corrected according to model specifications in the file header.

Figure 1 (this response letter) illustrates the treatment by comparing PDB and BDB-derived B-factors for all four structures. In the BDB, the site-specific pattern is maintained and the amplitude of fluctuations among the sites appears more consistent. We like to add that we do not draw conclusions based on these data; experimental B-factors are provided only to support our MD-derived atom mobilities.

Figure 1. Comparison of isotropic B-factors deposited in PDB and BDB data banks.

We simplified the corresponding sentence in the legend of Supplementary Fig. 21 (SI, lines 718-721):

“Full isotropic B-factors potentially reflect the mobility of the C α atoms of corresponding BCNK junctions in the protein backbone (mature numbering; Uniprot P24394-1). Values were taken from the databank of PDB files with consistent B-factors (BDB)³⁶ for the published crystal structures (PDB: 1IAR, 3BPL, 3BPO, 3BPN)^{14,35}.”

Fig S22:

color-ramp from green to magenta: only green spheres can be seen.

Response: This was a result of the specific angle under which the structures are shown. However, we noticed that the catchbox analysis providing the basis for the displayed contact area is introduced in the text just after calling Fig. S22. Therefore, the spheres and the corresponding text in the legend have been removed.

SI:

The description of how the charges were derived given in the reponse should be put in the README in the SI.

Response: The mentioned step-by-step description of how partial charges (and corresponding force field parameters) were derived has been included in the Supplementary toolkit (SI_comp_bio/a, README_BCNK_PARAMETERIZATION file).

It would be more useful if (instead of large movie files) the authors provided pymol/vmd sessions for the structures and deposited the trajectories to make them readily available to the reader.

Response: We understand the request and are in full compliance with transparency and accessibility of data. However, to balance efforts, we would like to prefer offering raw data like trajectories, parmtop/pdb/restart files etc. on demand. A single trajectory file consumes on the order of 13 GB of disk space, therefore too much to serve as a supplementary file. Efforts to establish a permanent repository appears costly considering the role of trajectories for this manuscript. As a reminder, our solvation free energy calculations are based on average structures that are already comprehensively included in the source data. Thus, the actual trajectories are not relevant to follow our conclusions. Nonetheless, for broadest coverage of also the non-specialist reader, we have prepared movies in standard mp4 format to provide an easy-to-follow picture of IL-4R α dynamics. Moreover, detailed instructions for straightforward reproduction of each of the simulations are contained in the Supplementary toolkit (SI_comp_bio.zip) so that on decent hardware a particular system can be fully reproduced from scratch within about 30 hours computing time. As such, we believe the scientific community is served with convenient means for looking deeper into the simulations and translate them to systems of interest.

Reviewer #2 (Remarks to the Author):

As mentioned in my original report, this is a very interesting manuscript that brings together cross-correlation autocorrelation analysis with confocal imaging to provide novel information on the efficiency of bioorthogonal labelling of particular amino acids on the extracellular elements of a single membrane-passing protein. As a consequence, new information is provided on the conformations adopted in response to ligands. I cannot comment on the molecular dynamic analysis and have confined my comments to the general approach used to investigate ligand-dependent receptor states.

The authors have adequately dealt with the major points that I raised previously. The provision of actual concentrations in Figure 2c and d have improved the understanding of the manuscript greatly. I have no other concerns. The changes to the Supplementary figures are also welcome.

Response: We thank the reviewer for the positive evaluation of our revised manuscript.

Reviewer #3 (Remarks to the Author):

The revised manuscript by F. Steiert and co-workers addresses my original comments. My major concern was the poor scholarly presentation of the otherwise very interesting results. This has

been resolved by including a discussion part that briefly summarizes the major findings of the work and is written in a more general language that is now much more enjoyable to read. I find that the additional changes improved the overall quality of the work and I do not have any further comments, suggestions or requests. Congrats to all the authors for a nice work.

Response: We highly appreciate the positive feedback by the reviewer.

Reviewers' Comments:

Reviewer #4:

Remarks to the Author:

The overall paper is intriguing and presents a novel approach to examining proteins in vivo. The simulation work suffers from the following problems which should be addressed:

(1) No indication of the statistical accuracy of the simulation results is given. This should be performed for all simulation-derived quantities. Normally in the analysis of structural or dynamical quantities several trajectories are performed and statistical analysis performed over these. Alternatively, in the absence of multiple trajectories single long trajectories may be performed and uncertainties quantified using a time dependence analysis.

(2) There is an interesting result correlating the PB solvation free energy with the click efficiency (CE). The better a particular receptor structure is stabilized by the solvent the less reactive the BCNK moiety. However, the likelihood that, despite the correlation, the general solvation free energy of a protein, which is averaged over a large surface area most of which is distant from the reaction site, directly influences the reaction rate seems low. Note that indirect effects of electrostatics via structure and dynamics cannot be postulated unless these changes are shown to directly be visible in the simulations and here they are not. Rather, this calls for an analysis of differences of the average electrostatic potential at the reactive moiety itself rather over the whole protein.

POINT-BY-POINT RESPONSE TO REVIEWER COMMENTS

Manuscript-ID: NCOMMS-22-37432C

Reviewer #1 (Remarks to the Author):

The overall paper is intriguing and presents a novel approach to examining proteins in vivo. The simulation work suffers from the following problems which should be addressed:

(1) No indication of the statistical accuracy of the simulation results is given. This should be performed for all simulation-derived quantities. Normally in the analysis of structural or dynamical quantities several trajectories are performed and statistical analysis performed over these. Alternatively, in the absence of multiple trajectories single long trajectories may be performed and uncertainties quantified using a time dependence analysis.

Response: The reviewer requested to elaborate on the statistical significance of MD-derived properties. This is particularly relevant for input structures of PB calculations, which up to now have solely been determined from single average structures of the entire MD trajectory. In this revision, 10 individual sub-trajectories were formed (10 runs a 25 ns) for which solvation free energies were computed in identical fashion as reported in the manuscript. Individual partial results were averaged and, consistent with experimental data, statistically characterized by the standard error of the mean. Following this procedure, SFE errors were of the same order of magnitude as those of the experimental click efficiencies (revised Fig. 5b and Figure 1a, response letter). Thus, all our previous arguments can be upheld and no major changes have been identified. This was largely anticipated as a predominant configuration in a particular sub-trajectory would increase its weight in the overall average formed from summing up the complete MD trajectory. Again, solvent polarization, ΔG^{pol} , is clearly demonstrated to be the key term with ΔG^{cav} and ΔG^{disp} largely canceling each other at significantly reduced overall variation (revised Supplementary Table 6, line 845 in the SI). Statistical variations are of comparable magnitude in each of the considered molecular systems bearing BCNK at different positions, hence making a direct comparison of mean values a reasonable choice of analysis.

To enhance confidence in the calculated SFE values, we additionally carried out frame-wise PB calculations for each of the 10,000 snapshots in the trajectory of all the 8 receptor mutants under consideration. The brute-force approach allowed tracking of standard deviations and the standard errors of the mean along the trajectory. The pattern of site-specific variations remained stable (Figure 1b and 1c, response letter). Averaging SFE-values for a subset of consecutive frames (~1,000) corresponding to the length of the partial sub-trajectories (25 ns) as described above, reproduced fairly well the statistics and error bounds (Figure 1c, response letter). Owing to the usage of structural snapshots directly from the MD trajectory without further minimization, there was a sizeable shift in absolute values of ΔG^{solv} . Interestingly, the shift affects all structures to a comparable degree, within the set of 10,000 snapshots as well as between the different molecular

systems. Thus, based on statistical considerations, the correlation between experimental CE and modelling-based SFE is fully supported.

Figure 1. Correlation between experimental CE and SFE calculated from MD simulations. (a) SFE values derived from 10 sub-trajectories (25 ns) after averaging and minimization as now shown in the revised manuscript (Fig. 5b). (b) Evaluation of individual frames of the trajectories or (c) averaged along ~1,000 consecutive frames. Note that averaging frames in (c) reproduced the magnitude of error bounds in (a). However, omitting minimization leads to a noticeable shift in SFE values (b and c). Error bars represent SD (grey) and SEM (black).

(2) There is an interesting result correlating the PB solvation free energy with the click efficiency (CE). The better a particular receptor structure is stabilized by the solvent the less reactive the BCNK moiety. However, the likelihood that, despite the correlation, the general solvation free energy of a protein, which is averaged over a large surface area most of which is distant from the reaction site, directly influences the reaction rate seems low. Note that indirect effects of electrostatics via structure and dynamics cannot be postulated unless these changes are shown to directly be visible in the simulations and here they are not. Rather, this calls for an analysis of differences of the average electrostatic potential at the reactive moiety itself rather over the whole protein.

Response: We share the reviewer's view that at every instant the nature of the local environment should determine the reaction rate at the terminal BCNK ring. We are therefore aware that the discovered correlation between solvation free energy (SFE) and the click efficiency does not reveal a physicochemical mechanism. To elaborate a physical model for reaction rates at the cell surface

is clearly beyond the scope of this manuscript. For comparative SFE calculations, the use of entire receptor structures bearing BCNK is technically straightforward. However, this procedure should not imply that all parts of the receptor surface contribute to the same extent, neither with respect to SFE nor local BCNK reactivity. Although the physicochemical driving force for the reaction remains hidden, we consider this finding an exciting and promising starting point for future investigations in that direction.

We like to inform the reviewer that we have searched intensively for local cues at the receptor that explain the measured site-specific variations of the reaction yield. Starting from nearest neighbors in the primary sequence and in crystal structures, it soon became clear that we needed to look closely at receptor dynamics and evaluate the mobility of BCNK at the receptor surface. Subsequently, MD simulations of BCNK-bearing receptor mutants revealed that the contact area can be unexpectedly large and diverse among the receptor mutants (Supplementary Figs 24-26). To evaluate the trajectories (250 ns) quantitatively, the catchbox analysis was developed (Fig. 4). Based on the comprehensive list of receptor atoms making contact with BCNK, we challenged the data set in search of correlations with local properties at the reactive moiety: particular amino acids, types of amino acids, types of atoms, hydrophobicity, size of the contact area, conformation and mobility of the BCNK side chain including the terminal ring — without success. Some of these intuitive approaches are mentioned in the text, others have been omitted for the sake of brevity. One of these were our preliminary attempts to analyze the local electrostatic potential (ESP) with APBS in PyMOL (neglecting BCNK).

To follow up on the reviewer's suggestion, we revisited this issue and illustrated the ESP surface of average structures in which BCNK was included. This required:

- 1) parametrization of the non-standard residue BCNK,
- 2) selection of target points for explicit calculation of ESPs following frequently detected atoms in the catchbox and corresponding points on the molecular surface (the latter requiring a reverse linkage of surface dots to solute atoms), and
- 3) selection of target points for ESP calculation of BCNK-related surface dots (again linking surface dots to specific atoms of the solute).

The contact area on the receptor surface features for all receptor mutants a quite large and heterogeneous charge distribution (Figure 2, response letter; additional movies for a better 3D-imagination of the ESP maps are supplied as “Auxiliary Movies 1–7 for Reviewer #4”). As we have seen in our preliminary attempts, the position of negatively or positively charged hotspots at varying distances to the reactive BCNK moiety are altogether inconclusive. The apolar BCNK is energetically poised to avoid such hotspots and locate in undefined regions in between. Thus, unlike modeling attempts for ligand binding, where the charge distribution of the ligand is precisely known, we have to deal with a highly mobile hydrophobic moiety in BCNK that cannot be matched to charge patterns at the receptor surface.

At the current stage we can conclude the following: SFE values depend sensibly on the position of the reactive BCNK moiety relative to the receptor surface. Therefore, the position of the reactive

moiety in the average structure can be considered to be both productive and representative for the reaction yield. Notably, the observed correlation between click efficiency and SFE is based on the average position of the reactive BCNK moiety alone, therefore neglects entirely the position of the charged fluorescent tetrazine as a reaction partner (Supplementary Fig. 25). As the position of charged reaction partners within the heterogeneous ESP landscape is irrelevant, the picture of charge-mediated docking of reactants prior to clicking is not supported.

Figure 2. Electrostatic potential (ESP) maps of BCNK-bearing receptor mutants. Average structures of receptor mutants (PyMOL, ‘cartoon’ representation, palegreen) containing the BCNK side chain (‘sticks’ representation, orange) were overlaid with spheres representing either surface elements of BCNK or surface elements of the receptor neighborhood that were in contact with the BCNK terminal ring during the MD simulation (catchbox analysis). The spheres are color coded for the ESP at that particular surface element.

In the current manuscript, we try to be cautious with speculations on this issue; also, as a result of previous comments by other reviewers. To discuss the significance of the observed correlation (SFE vs. CE), we added in the main text (Line 392): “Although the physicochemical driving force for the reaction rate remains elusive, this correlation suggests that the iEDDAC reaction is driven by a mechanism that is critically conditioned by the surrounding solvent. The average position of the reactive BCNK moiety can be considered to be both productive and representative for the reaction yield. However, as the position of the charged fluorescent tetrazine as a reaction partner is neglected in the observed correlation, a mechanism that involves charge-mediated docking of reactants prior to conjugation is not supported.”

Reviewers' Comments:

Reviewer #4:

Remarks to the Author:

The authors have done a good job here responding to the comments. The manuscript should be published.